# Ubiquilin1 promotes antigen-receptor mediated proliferation by eliminating mislocalized mitochondrial proteins

Alexandra M Whiteley[1,2], Miguel A Prado[2], Ivan Peng[3], Alexander R Abbas[4], Benjamin Haley[5], Joao A Paulo[2], Mike Reichelt[6], Anand Katakam[6], Meredith Sagolla[6], Zora Modrusan[7], Dong Yun Lee[1†], Merone Roose-Girma[7], Donald S Kirkpatrick[7], Brent S McKenzie[3], Steven P Gygi[2], Daniel Finley[2], Eric J Brown[1]*

[1]Department of Infectious Disease, Genentech, South San Francisco, United States; [2]Department of Cell Biology, Harvard Medical School, Boston, United States; [3]Department of Translational Immunology, Genentech, South San Francisco, United States; [4]Department of Bioinformatics, Genentech, South San Francisco, United States; [5]Department of Molecular Biology, Genentech, South San Francisco, United States; [6]Department of Pathology, Genentech, South San Francisco, United States; [7]Department of Microchemistry, Proteomics, and Lipidomics, Genentech, South San Francisco, United States

*For correspondence:
ericbr@gene.com

Present address: †Manufacturing Sciences and Technology, Genentech, Vacaville, United States

**Abstract** Ubiquilins (Ubqlns) are a family of ubiquitin receptors that promote the delivery of hydrophobic and aggregated ubiquitinated proteins to the proteasome for degradation. We carried out a proteomic analysis of a B cell lymphoma-derived cell line, BJAB, that requires UBQLN1 for survival to identify UBQLN1 client proteins. When UBQLN1 expression was acutely inhibited, 120 mitochondrial proteins were enriched in the cytoplasm, suggesting that the accumulation of mitochondrial client proteins in the absence of UBQLN1 is cytostatic. Using a $Ubqln1^{-/-}$ mouse strain, we found that B cell receptor (BCR) ligation of $Ubqln1^{-/-}$ B cells led to a defect in cell cycle entry. As in BJAB cells, mitochondrial proteins accumulated in BCR-stimulated cells, leading to protein synthesis inhibition and cell cycle block. Thus, UBQLN1 plays an important role in clearing mislocalized mitochondrial proteins upon cell stimulation, and its absence leads to suppression of protein synthesis and cell cycle arrest.

DOI: https://doi.org/10.7554/eLife.26435.001

## Introduction

The maintenance of optimal intracellular protein concentrations through regulated degradation is central to cellular homeostasis, and the proteasome mediates this process by recognizing ubiquitinated substrates and translocating them into its proteolytic core for degradation. Additional layers of control are added by the expression of ubiquitin receptors, which bind to the proteasome and regulate the degradation of client proteins. Ubiquilins (Ubqlns) belong to a family of UBL/UBA proteins that link proteasomes with ubiquitinated protein cargo through a ubiquitin-like domain (UBL) that binds to the proteasome (*Elsasser et al., 2002*; *Elsasser et al., 2004*; *Shi et al., 2016*) and a ubiquitin-binding domain (UBA) that is capable of binding mono- and polyubiquitin chains (*Funakoshi et al., 2002*; *Kleijnen et al., 2000*; *Ko et al., 2004*; *Zhang et al., 2008*; *Zhang et al., 2009*). While the UBA domain of Ubqlns appears to show limited specificity for particular ubiquitin linkages (*Zhang et al., 2008*), Ubqlns have one or more additional structural elements (*Kaye et al.,*

*2000*; *Itakura et al., 2016*; *Ford and Monteiro, 2006*) that may direct the specificity of Ubqln-mediated protein degradation toward distinct client proteins. In yeast, the Ubqln ortholog *DSK2* is required for proteasome-mediated degradation of a subset of polyubiquitinated proteins (*Shi et al., 2016*; *Funakoshi et al., 2002*; *Verma et al., 2004*; *Elsasser and Finley, 2005*; *Lim et al., 2009*). Consistent with these studies in yeast, work on mammalian Ubqln proteins suggests a role in proteasomal degradation (*Kleijnen et al., 2000*; *Itakura et al., 2016*; *Ford and Monteiro, 2006*; *Hjerpe et al., 2016*; *Chang and Monteiro, 2015*; *Stieren et al., 2011*).

Although the hypothesis that Ubqlns function to shuttle-specific proteins directly to protein degradation machinery has experimental support, the full repertoire of Ubqln client proteins, as well as the circumstances under which Ubqlns are required for their degradation, remain poorly understood. The prevailing theory of Ubqln function is that they assist in the degradation of aggregated or misfolded proteins through UBA domain and client ubiquitin chain interactions, with client protein specificity conferred by the central portions of Ubqln (*Itakura et al., 2016*; *Hjerpe et al., 2016*). However, a comprehensive accounting for the proteins dependent on Ubqlns for their degradation, and of those which are pathological upon their accumulation, is lacking. This problem has previously been approached from two directions: by using ubiquitin as a marker of aggregated protein to identify the sensitive tissues and cells of Ubqln-deficient in vivo models, and by studying proteins that are known to aggregate and cause pathology in Ubqln-deficient model systems (*Hjerpe et al., 2016*; *Stieren et al., 2011*; *El Ayadi et al., 2012*; *Ford and Monteiro, 2006*; *Picher-Martel et al., 2015*). By employing multiplexed proteomics on cells sensitive to UBQLN1 depletion, we now are able to determine the proteins that are dependent on UBQLN1 for their elimination in an unbiased fashion.

Recently, Itakura et al. (*Itakura et al., 2016*) reported that UBQLN1 binds to a variety of mitochondrial transmembrane proteins and is necessary for the delivery of mislocalized mitochondrial proteins for proteasomal degradation. Membrane proteins that fail to be properly inserted into mitochondria due to defects or inefficiencies in the mitochondrial protein translocation machinery require UBQLN1 for their delivery to proteasomes. Under these conditions, the central portion of UBQLN1 is required to bind hydrophobic domains of mitochondrial proteins in order to promote their degradation via the proteasome.

However, Ubqln function is not limited to cytosolic aggregates and mitochondrial proteins: UBQLN1 also binds to the ER membrane protein Erasin, a membrane component of the ER-associated degradation (ERAD) pathway (*Lim et al., 2009*), and Dsk2 binds to the E4 ubiquitin ligase UFD2, which transfers client proteins from CDC48 to the proteasome (*Medicherla et al., 2004*; *Richly et al., 2005*; *Liu et al., 2009*; *Hänzelmann et al., 2010*). UBQLN4 binds to (*Lee et al., 2013a*) and colocalizes with (*Rothenberg et al., 2010*) LC3, and the loss of Ubqln results in sensitivity to starvation (*N'Diaye et al., 2009b*). In these systems, it appears that some Ubqlns may deliver ubiquitinated proteins to growing autophagosomes (*N'Diaye et al., 2009a*).

In addition to direct regulation of protein degradation via proteasomes and autophagy, Ubqlns have been indirectly linked to proteome stability through interactions with mTORc1 (*Wu et al., 2002*) and the calcium channel ORAI1 (30). Ubqlns also influence the signaling activity of TLRs (*Biswas et al., 2011*), and GPCRs (*N'Diaye et al., 2008*). In these situations, it remains unclear whether the signaling activity of Ubqlns reflects an indirect effect of their role as proteasome receptors, or whether they play a distinct and non-proteolytic role in signal transduction.

Despite recent insight into UBQLN2's role in diseases such as ALS-FTD (*Hjerpe et al., 2016*; *Chang and Monteiro, 2015*; *Deng et al., 2011*; *Gorrie et al., 2014*; *Le et al., 2016*), there is little understanding of how Ubqlns function in normal mammalian cell biology or physiology. To investigate this question, we examined *UBQLN1*, the most broadly expressed member of the Ubqln family (*Marín, 2014*). Using a combination of targeted approaches and unbiased mass spectrometry analysis, we found that UBQLN1-deficient cells accumulate mislocalized mitochondrial proteins in the cytosol, leading to a loss in viability. To understand the importance of this function in normal physiology, we created and studied a *Ubqln1*-deficient mouse. In the absence of *Ubqln1*, there were abnormalities in B lymphocytes, including a decreased number of B1a B cells, which uniquely rely on BCR signaling for their self-renewal. *Ubqln1*-deficient B cells failed to proliferate in response to BCR ligation, a stimulus that uniquely depolarizes mitochondria, and accumulated mitochondrial proteins compared to their WT counterparts. $Ubqln1^{-/-}$ B cells also did not increase the cell cycle proteins Cyclin D2 and Cyclin D3 normally after BCR ligation, which reflected a failure to upregulate protein

synthesis, while protein synthesis downstream of other stimuli was still strongly upregulated. We hypothesize that a toxic combination of mitochondrial depolarization and defective protein degradation leads to the failure of protein synthesis upon stimulation of B cells via the BCR. Overall, these findings suggest that Ubqlns can serve as indirect regulators of protein synthesis and cell cycle progression, and highlight the sensitivity of stimulated cells to proteomes that have been perturbed as a consequence of mitochondrial depolarization.

## Results

### Loss of *UBQLN1* leads to cytosolic accumulation of mitochondrial proteins

To generate a comprehensive understanding of the *UBQLN1*-regulated proteome, we used multiplexed proteomics of UBQLN1-depleted cells to identify proteins accumulated in the absence of *UBQLN1*. Doxycycline-inducible *UBQLN1*-targeted small hairpin RNAs were stably introduced into BJAB, a germinal center B cell diffuse large cell lymphoma (GCB DLBCL) cell line characterized by high expression of mitochondrial proteins involved in oxidative phosphorylation, including ATP5G1 (*Monti et al., 2005*; *Brien et al., 2007*). BJABs also exhibit tonic BCR signaling (*Young and Staudt, 2013*; *Davis et al., 2010*), high AKT activity (*Pfeifer et al., 2013*) and are sensitive to the loss of Cyclin D (*Schmitz et al., 2012*) and c-MYC (*Pfeifer et al., 2013*). After 48 hr of culture with doxycycline, BJABs expressing the *UBQLN1* shRNA construct had lost >85% of UBQLN1 protein and roughly 35% of UBQLN2 protein (*Figure 1A*). Unlike other tested cell lines (*Itakura et al., 2016*; *N'Diaye et al., 2009b*), BJAB survival and proliferation were profoundly sensitive to the loss of UBQLN1 (*Figure 1B–D*). Loss of proliferation was due to depletion of UBQLN1, since the loss in cell proliferation was strongly correlated with the amount of UBQLN1 depletion for each shRNA tested (*Figure 1—figure supplement 1* ), and the loss of proliferation was rapidly reversed upon removal of doxycycline from the culture (*Figure 1E*).

To determine the effect of UBQLN1 depletion on the proteome, we performed a TMT-MS3 analysis on isolated cytosol from BJAB cells expressing either *UBQLN1* or control shRNA. After 48 hr of dox induction, cytosolic proteins were isolated from cells and prepared for mass spectrometry. Then, triplicate samples from *UBQLN1* or control shRNA-expressing cells of three independent experiments were labeled with TMT reagents to allow for 6TMT multiplexing of the peptide libraries in one MS run. More than 5100 proteins were quantified, 1284 of which were significantly altered ($p < 0.05$ of a Student's T test) between normal and Ubqln1-depleted cells (*Figure 1—source data 1*). Confirming our results from *Figure 1A*, we found that both UBQLN1 and UBQLN2 were significantly depleted from BJABs expressing *UBQLN1* shRNA (*Figure 1C*).

DAVID pathway analysis (*Huang et al., 2009*) for the cellular compartment of proteins enriched by at least 2-fold upon UBQLN1 depletion (92 proteins) demonstrated a strong mitochondrial protein signature in the cytosol of UBQLN1-depleted cells, which was validated by cross-referencing our entire dataset with proteins annotated by Mitocarta 2.0 (*Calvo et al., 2016*) (*Figure 1F* and *Table 1*). All detected proteins were scored for their relative hydropathy using the GRAVY (GRand AVerage of HYdropathy) calculation, which divides the sum Kyte Doolittle hydropathy score for each amino acid by the length of the protein (*Kyte and Doolittle, 1982*). The GRAVY score for each protein was then plotted against the protein's abundance in normal and UBQLN1-knockdown cytosol (see Materials and methods for details). We found an enrichment of hydrophobic proteins among those that accumulated in UBQLN1-depleted cytosol, many of which are known to be localized to the mitochondrion in wild-type cells (*Figure 1G* and *Figure 1—source data 1*). None of the 12 mitochondrially encoded mitochondrial proteins were quantified in our dataset (*Figure 1—source data 1*), suggesting that the observed enrichment of mitochondrial proteins in the cytosol was not due to rupture of damaged mitochondria or to opening of the mitochondrial transition pore. Thus, mislocalized nucleus-encoded mitochondrial proteins make up the dominant fraction of UBQLN1 substrates in this system, consistent with the model presented by *Itakura et al. (2016)*.

To confirm these findings, we examined the localization of ATP5G1, the membrane component of the mitochondrial ATP synthase, which is known to require interaction with Ubqlns for its cytosolic degradation (*Itakura et al., 2016*) but was not quantified in our proteomics study. Immature ATP5G1 runs at approximately 17 kDa; when the protein is successfully inserted into mitochondria,

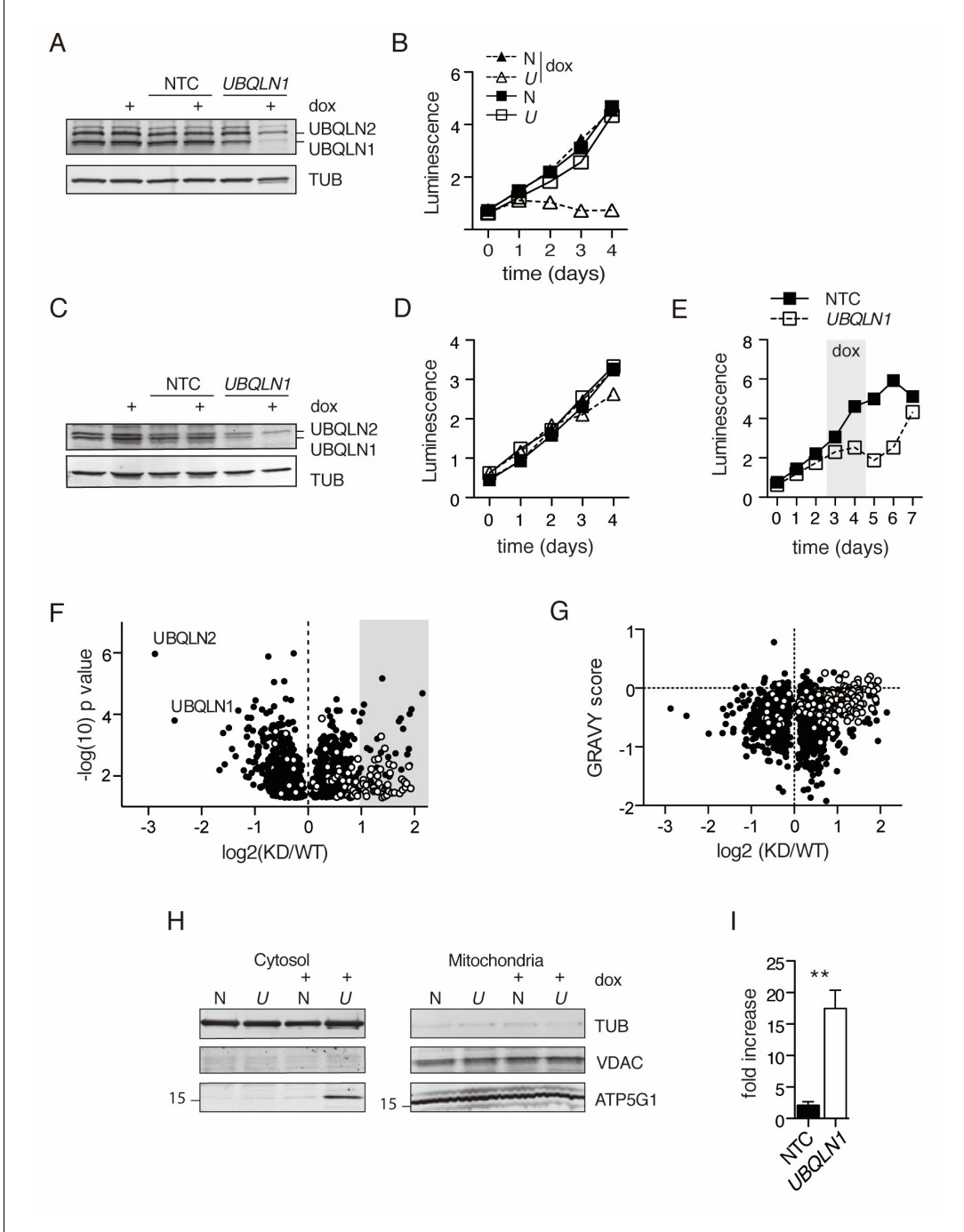

**Figure 1.** Mitochondrial proteins accumulate in the cytosol in the absence of UBQLN1. (**A**) Uninfected BJABs, or cells infected with a lentiviral construct containing non-targeted control (NTC) or *UBQLN1*- shRNA (construct #1 from *Figure 1—figure supplement 1*) were incubated with 100 ng/mL doxycycline (+dox) or vehicle for 72 hr, following which UBQLN1 and Tubulin were detected by western blot. (**B**) BJAB cells stably transfected with *UBQLN1*- (*U*, open symbols) or an NTC-shRNA (N, filled symbols) were incubated with doxycycline (triangles) or vehicle (squares) for 4 days. Viable cells were measured with the Cell Titer Glo (Promega) assay in duplicate with mean ±SEM, and luminescence quantified as arbitrary units. One representative experiment of two is shown. (**C**) Untransfected HeLa cells, or cells stably transfected with NTC- or *UBQLN1* shRNA were treated similarly to BJABs and western blotted as in *Figure 1A*. (**D**) Viable cells were measured as in (*Figure 1B*) for HeLa cells. (**E**) BJAB cells containing either an NTC- or *UBQLN1* shRNA construct (#1) were cultured in the presence of 100 ng/mL doxycycline starting at day 2 and then washed to remove doxycycline (grey-shaded box). The number of viable cells was estimated by CellTiter Glo as arbitrary units of luminescence in duplicate wells. Each line represents the average of duplicate wells with mean ± SEM. (**F**) Isolated cytosolic protein from NTC- or *UBQLN1*- shRNA expressing BJABs was labeled in triplicate with TMT and analyzed by mass spectrometry. Shown are the roughly 1300 significantly altered (p<0.05) proteins and their relative enrichment

*Figure 1 continued on next page*

*Figure 1 continued*

in UBQLN1-depleted cytosol. Mitochondrial proteins, as listed in Mitocarta 2.0 (*Calvo et al., 2016*), are shown in white. Grey box highlights proteins that were at least twofold accumulated in KD cytosol (92 proteins), which were used for DAVID pathway analysis. (G) Proteins identified in (F) were plotted according to their GRAVY score against their relative enrichment in UBQLN1-depleted cytosol. As in (F), mitochondrial proteins are shown in white. (H) Cellular fractionation of BJAB cells after 48 hr with 100 ng/mL doxycycline. Cells were fractionated according to Materials and Methods. Samples were blotted for mitochondrial proteins VDAC and ATP5G1, as well as for Tubulin. ATP5G1 ran at 17 kDa and thus represents the precursor form of the protein before mitochondrial insertion. N: Non-targeting shRNA construct. *U: UBQLN1* shRNA construct. Shown is one representative experiment of three. (I) Quantification of ATP5G1 in cytosolic fractions of BJABs from three independent experiments. ATP5G1 was normalized to Tubulin in each sample. Data shown are mean ±SEM, and significance was determined with an unpaired Student's T test.

DOI: https://doi.org/10.7554/eLife.26435.002

The following source data and figure supplement are available for figure 1:

**Source data 1.** TMT MS3 results for BJAB cytosol.
DOI: https://doi.org/10.7554/eLife.26435.004
**Figure supplement 1.** Selection of UBQLN1 shRNAs.
DOI: https://doi.org/10.7554/eLife.26435.003

the mitochondrial targeting sequence (MTS) is clipped, leading to a smaller, 11 kDa protein. Our antibody recognized uncleaved, precursor ATP5G1 much more readily than the mature, cleaved form; therefore, the ATP5G1 band visualized in mitochondria represents protein in the process of insertion. There was a major increase of unprocessed ATP5G1 in the cytosol, but not mitochondria, of BJABs after induction of *UBQLN1* shRNA (*Figure 1H*). In the cytosol, we found approximately 10- to 20-fold higher levels of ATP5G1 following *UBQLN1* depletion (*Figure 1I*).

## $Ubqln1^{-/-}$ mice have altered B cell numbers

The unique sensitivity of BJAB cells to UBQLN1 loss led us to hypothesize that B cells, or their proliferation in response to B cell stimulation, may require UBQLN1. To understand the significance of *UBQLN1* loss in normal B cell physiology, we generated *Ubqln1* knockout (-/-) mice by deleting exon 2 (*Figure 2A*). Loss of *Ubqln1* mRNA and protein was confirmed by RT-PCR and western blot (*Figure 2B–C*). Mice were generally healthy, and we observed no gross anatomical abnormalities. However, immunophenotyping of $Ubqln1^{-/-}$ mice demonstrated a deficiency of peritoneal B1a cells (*Figure 2D*) despite normal levels of typical surface markers (*Figure 2—figure supplement 1A*). B1a cells are a subset of B1 cells that express CD5 and, like all B1 cells, they uniquely self-renew in the peritoneum by a pathway dependent on BCR signaling (*Lam and Rajewsky, 1999*; *Hayakawa et al., 1985*). Depletion of BCR signaling components, including proximal regulators such as PKCβ (*Leitges et al., 1996*) as well as distal regulators like Cyclin D2 (*Solvason et al., 2000*), can cause a decrease in the B1a population, while sparing B2 cells, whereas depletion of negative regulators of BCR signaling causes an increase in B1a cells (*Berland and Wortis, 2002*). $Ubqln1^{-/-}$ mice had normal numbers of total splenic B-cells (*Figure 2E–F*), normal proportions of transitional B-cell populations (*Figure 2—figure supplement 1B*), and normal serum IgM and IgG levels (*Figure 2—figure*

**Table 1.** GO-term CC cluster analysis of proteins increased by *UBQLN1* loss in isolated BJAB cytosol.

DAVID pathway analysis was used on a dataset of proteins enriched at least 2-fold with p value >< 0.05 (92 proteins) in isolated BJAB cytosol upon loss of UBQLN1 via shRNA induction using the background of all identified peptides from the MS run (approx. 5000 proteins). GO-term Cellular Compartments were used for a cluster analysis; clusters with Enrichment Scores of > 1.3 (p=0.05 equivalent) were listed with their common term.

| Term | Enrichment Score |
| --- | --- |
| Mitochondria | 19.66 |
| Mitochondrial Membrane | 9.05 |
| Extracellular Vesicle | 3.34 |
| Membrane-bound organelle | 2.66 |

DOI: https://doi.org/10.7554/eLife.26435.005

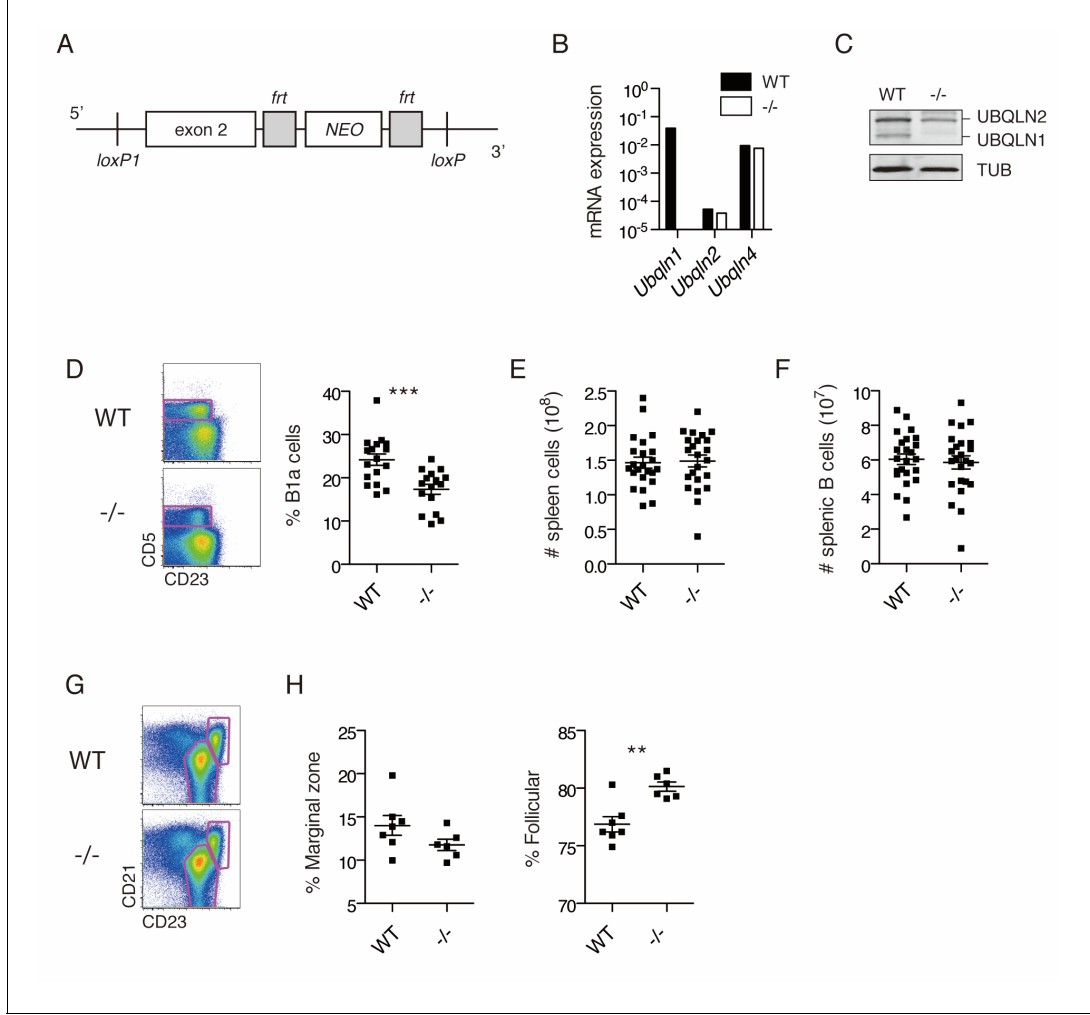

**Figure 2.** *Ubqln1^{-/-}* mice have a specific defect in peritoneal B cells. (**A**) *Ubqln1* knockout strategy. Exon 2 of *Ubqln1* was floxed and the *Neo* gene deleted in vitro by using Flp recombinase. Exon 2 was deleted by expression of CRE in ES cells. (**B**) mRNA expression of *Ubqlns1*, *2*, and 4 in isolated splenic B cells. Each *Ubqln* mRNA was normalized to *Gapdh. Ubqln1* appears negative due to deletion of the genomic region containing QPCR primer. (**C**) Protein levels of UBQLNs 1 and 2 in primary B cells by western blot. No truncated UBQLN1 protein product could be detected by polyclonal antibody. (**D**) Peritoneal B1a cells are decreased as a proportion of total peritoneal B cells in *Ubqln1^{-/-}* mice. B1a cells were identified as a proportion of total B cells in the peritoneum from mice $\geq$ 8 weeks of age. On the left are representative dot plots of B cells (CD19$^+$ FSC$^{lo}$ cells) showing B1a cell gating strategy. Graph on the right depicts % of peritoneal B cells that are B1a. Each point represents one mouse. (**E**) Total spleen cells in WT or *Ubqln1^{-/-}* littermate mice at least 8 weeks of age. (**F**) Total number of splenic B cells in WT or *Ubqln1^{-/-}* littermate mice at least 8 weeks of age. (**G**) Representative gating strategy for identification of Marginal zone (top right gate) and Follicular B cells (bottom gate) of the spleen. Shown are live B cells (CD19$^+$, FSC$^{lo}$). (**H**) Quantification of Marginal zone (CD21$^+$, CD23$^{int}$) and Follicular B cells (CD21$^{int}$, CD23$^+$) as a proportion of total splenic B cells. For (D, E, F, and H), data shown are mean ±SEM and significance was determined with a paired T-test.

DOI: https://doi.org/10.7554/eLife.26435.006

The following figure supplement is available for figure 2:

**Figure supplement 1.** Additional phenotyping of *Ubqln1^{-/-}* mice.
DOI: https://doi.org/10.7554/eLife.26435.007

*supplement 1C*). However, we did observe a modest dysregulation of splenic B-cell populations from the marginal zone and follicles (***Figure 2G–H***).

## *Ubqln1* is required for cell cycle entry downstream of BCR stimulation

To determine whether the decreased B1a cell number in *Ubqln1^{-/-}* mice was related to a defective response to BCR ligation, splenic B cells were activated in vitro using either BCR ligation via anti-IgM crosslinking or Toll-like receptor (TLR) ligation with lipopolysaccharide (LPS). After 72 hr in

culture with anti-IgM, approximately 27% of WT B cells were alive, whereas only ~12% of $Ubqln1^{-/-}$ B cells were viable (*Figure 3A–B*). Furthermore, most wild-type cells divided more than once during this period, while the majority of $Ubqln1^{-/-}$ B cells failed to divide more than once (*Figure 3C*). Before committing to cell division, B cells activated via BCR crosslinking undergo a c-MYC-dependent increase in cell size and protein content (*DeFranco et al., 1985*; *Iritani and Eisenman, 1999*).

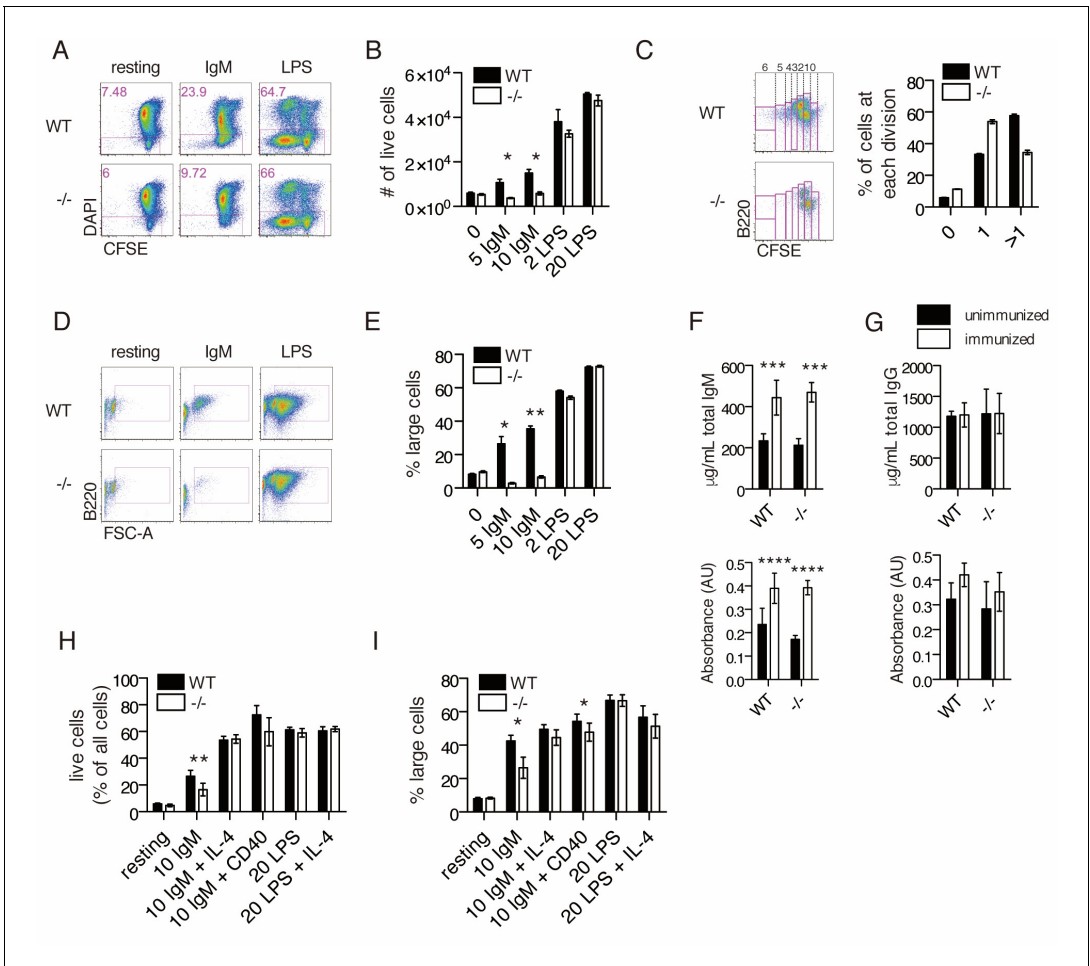

**Figure 3.** $Ubqln1^{-/-}$ B cells are defective in response to BCR stimulation. (**A**) Isolated splenic B cells from WT or $Ubqln1^{-/-}$ mice ≥8 weeks old were CFSE-stained and incubated with mitogen for 3 days. Shown are representative dot plots with live cells gated in pink. (**B**) Quantification of live cells in each well after 3 days of stimulation from one representative experiment of 6. Significance was determined with unpaired T tests. (**C**) CFSE dilution of IgM-stimulated WT and $Ubqln1^{-/-}$ cells treated as in (**A**) was analyzed by FACS, and fluorescence dilution converted to number of cell divisions. On the left are representative dot plots showing quantification of CFSE dilution. Number of cell divisions is demonstrated at top. Quantification of triplicate samples from the same experiment are shown at the right, by summing the populations within gates of cells that had divided at least once. (**D**) Live cells from (**A**) were plotted based on their size and B220 expression. (**E**) Quantification from triplicate samples in (**D**). One representative experiment of 6 is shown. Significance was determined with unpaired T tests. (**F**) Serum IgM antibody titers from WT or $Ubqln1^{-/-}$ mice immunized intraperitoneally with TNP-ficoll. Top: total IgM. Bottom: TNP-specific IgM. (**G**) Serum IgG antibody titers as in (**H**). Top: total IgG. Bottom: TNP-specific IgG. For (**F–G**), experiment was performed three times with at least six mice per genotype. (**H–I**) WT and $Ubqln1^{-/-}$ B cells were incubated in vitro with anti-IgM with or without addition of costimulation with either 10 μg/mL CD40 antibody or 40 ng/mL IL-4 for 3 days, following which the percentage of live cells (**H**) or the percentage of large live cells (**I**) were enumerated. Shown in (**H–I**) is summary data from N = 6 paired littermates. Significance was determined by paired T-test.

DOI: https://doi.org/10.7554/eLife.26435.008

The following figure supplements are available for figure 3:

**Figure supplement 1.** No in vitro B cell defects in the absence of other Ubqlns.
DOI: https://doi.org/10.7554/eLife.26435.009
**Figure supplement 2.** BAFFR signaling is not affected in $Ubqln1^{-/-}$ B cells.
DOI: https://doi.org/10.7554/eLife.26435.010

This step was markedly defective in $Ubqln1^{-/-}$ cells in response to BCR stimulation (*Figure 3D–E*). These abnormalities were specific for BCR-initiated proliferation, as we found no defect in $Ubqln1^{-/-}$ B cells at the resting state or in response to LPS in comparison to WT B cells (*Figure 3A*). Defective proliferation in response to antigen receptor crosslinking was specific to *Ubqln1*, as neither *Ubqln2*- nor *Ubqln4*-deficient B cells showed any abnormality in response to either anti-IgM or LPS (*Figure 3—figure supplement 1A–B*).

To determine the effects of *Ubqln1* deficiency on in vivo immune responses, we challenged the mice with an intraperitoneal injection of TNP-ficoll, which crosslinks the BCR. 10 days after immunization, we collected blood for serum IgM and IgG ELISAs and compared with serum collected before immunization. For both WT and $Ubqln1^{-/-}$ mice, we saw a significant increase in both total and TNP-specific IgM, but no change in IgG levels (*Figure 3F–G*). BCR-driven development and proliferation can be enhanced by the addition of costimulatory signals, which could account for the relatively normal proportions of splenic B cells in $Ubqln1^{-/-}$ mice and the apparently normal TNP-ficoll immunization results despite significant in vitro proliferation defects. Indeed, the viability and cell size defects of $Ubqln1^{-/-}$ B cells were at least partially rescued following the addition of costimulation via either the cytokine IL-4 or anti-CD40 crosslinking together with anti-IgM (*Figure 3H–I*). We investigated whether BAFFR signaling could compensate for defective BCR signaling in vivo by stimulating isolated B cells with recombinant BAFF over a 3-day period, and found no *Ubqln1*-dependent differences (*Figure 3—figure supplement 2*).

Since antigen receptor-stimulated primary murine B cells lacking UBQLN1 mimicked the proliferation defect observed in BJAB cells depleted of UBQLN1, we performed an experiment to characterize the proteome of stimulated $Ubqln1^{-/-}$ cells. 10-plexed TMT analysis on whole-cell lysates of WT and $Ubqln1^{-/-}$ B cells at 24 hr post-BCR stimulation quantified roughly 8000 total proteins, 822 of which were significantly altered between WT and $Ubqln1^{-/-}$ (*Figure 4—source data 1*). Overall, we observed smaller differences in the proteomes of stimulated whole murine B cells as compared to isolated cytosol from BJABs (*Figure 4A*, *Figure 4—source data 1*), likely because whole-cell extracts are not able to resolve mislocalized proteins in the cytosol from those present in mitochondria. Although a clear mitochondrial signature in the enriched protein fraction by Mitocarta 2.0 was not found (*Figure 4—figure supplement 1*), a DAVID pathway analysis (*Huang et al., 2009*) of proteins enriched at least 1.25-fold in $Ubqln1^{-/-}$ cells demonstrated increased abundance of ER proteins and mitochondrial membrane proteins in the absence of *Ubqln1* (*Table 2*). Furthermore, in silico analysis demonstrated a similar enrichment of hydrophobic proteins in $Ubqln1^{-/-}$ cells as in UBQLN1-depleted BJAB (*Figure 4B*). Non-mitochondrial hydrophobic proteins enriched in $Ubqln1^{-/-}$ cells included many transmembrane and some ER-resident proteins, suggesting that there could be a broader accumulation of membrane proteins in the absence of *Ubqln1*. However, since these results were generated with whole-cell extracts, as opposed to cytosolic extracts in *Figure 1*, we cannot distinguish between the possibility that these proteins accumulate within their destination organelle or membrane as opposed to being terminally mislocalized in the cytosol. Taken together, both complementary, comprehensive proteomic analyses support the hypothesis that *Ubqln1* deficiency leads to accumulation of mitochondrial and hydrophobic proteins, especially in the cytosol, and especially in the context of cellular stimulation.

## BCR stimulation induces prolonged mitochondrial depolarization in a *Ubqln1*-independent fashion

The accumulation of mitochondrial proteins in $Ubqln1^{-/-}$ cells following stimulation led us to examine whether activated cells undergo distinct mitochondrial stresses. B cell activation via the BCR induced prolonged mitochondrial depolarization (*Figure 4C*), which was not seen with LPS (*Souvannavong et al., 2004*). BCR (IgM) and TLR (LPS) stimulation both induced mitochondrial expansion (*Figure 4D* and *Figure 4—figure supplement 2B*) and mitochondrial superoxide production (*Figure 4E*). No *Ubqln1*-dependent changes in these measurements of mitochondrial health were detected, nor any significant loss in mitochondrial output as measured by ATP production and oxygen consumption rate (*Figure 4—figure supplement 2B–C*). Thus, unlike LPS, BCR stimulation causes long-lasting mitochondrial depolarization.

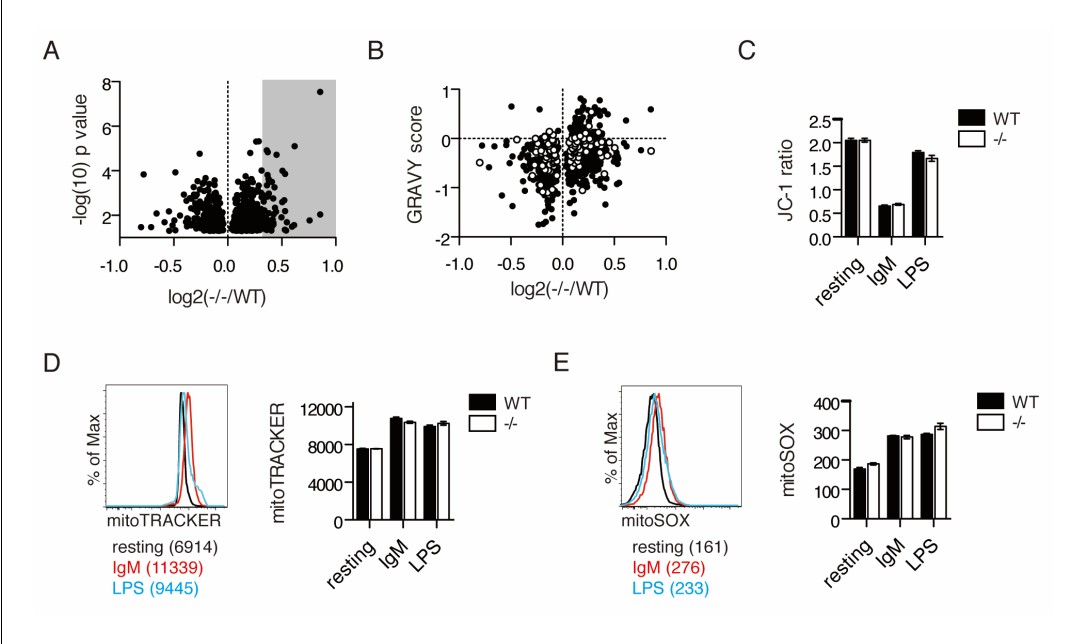

**Figure 4.** IgM stimulation causes *Ubqln1*-dependent changes in the proteome and depolarization of mitochondria. (A) Cell lysates from five samples each of WT or *Ubqln1*⁻/⁻ splenic B cells at 24 hr post-stimulation with 10 μg/mL anti-IgM were labeled with TMT 10-plex and analyzed by mass spectrometry. Shown are the roughly 800 annotated proteins that were significantly altered in *Ubqln1*⁻/⁻ cells compared to WT cells. Not shown is UBQLN1 at log2(-/-/WT)=−2.720. Grey shadow represents hits that were at least 1.25-fold enriched in *Ubqln1*⁻/⁻ lysate and used for DAVID pathway analysis. (B) As in *Figure 1*, GRAVY scores were calculated for proteins that were significantly altered between WT and *Ubqln1*⁻/⁻ samples and plotted against their relative enrichment in *Ubqln1*⁻/⁻ cells. As in *Figure 1*, mitochondrial proteins are shown in white. (C) Mitochondrial polarity of cells following stimulation was assessed with JC-1 dye. B cells cultured in triplicate in complete media or stimulated with anti-IgM or LPS for 24 hr were stained with JC-1 and the ratio of FITC to PE fluorescence o JC-1 dye was measured by flow cytometry. (D) Mitochondrial content of stimulated WT and *Ubqln1*⁻/⁻ cells as measured with Mitotracker dye as in (C). On the left is a representative histogram showing Mitotracker staining, with Mean Fluorescence Intensity of one representative sample from each stimulation condition of WT cells shown in parentheses. (E) Mitochondrial ROX content as measured by Mitosox dye with cells as in (C–D). On the left is a representative histogram showing Mitosox staining, with Mean Fluorescence Intensity of one representative sample from each stimulation condition of WT cells shown in parentheses. For (C–E), data shown are mean ± SEM of biological triplicates from one of three independent experiments each. Significance was determined with an unpaired T test.

DOI: https://doi.org/10.7554/eLife.26435.011

The following source data and figure supplements are available for figure 4:

**Source data 1.** TMT MS3 results for mouse B cell lysate.
DOI: https://doi.org/10.7554/eLife.26435.014
**Figure supplement 1.** Mitocarta visualization of B-cell proteome.
DOI: https://doi.org/10.7554/eLife.26435.012
**Figure supplement 2.** Additional mitochondrial tests show no effect of *Ubqln1* loss.
DOI: https://doi.org/10.7554/eLife.26435.013

## *Ubqln1*⁻/⁻ B cells fail to enter cell cycle following BCR stimulation

Despite normal mitochondrial responses to BCR stimulation, *Ubqln1*⁻/⁻ B cells accumulate mitochondrial proteins and fail to proliferate normally. To understand how the accumulation of mislocalized hydrophobic proteins might relate to the defect in B-cell proliferation, we investigated cell signaling downstream of BCR ligation in *Ubqln1*⁻/⁻ cells. BCR stimulation is a unique mitogen because it causes apoptosis in addition to proliferation (*Donjerković and Scott, 2000*). To determine the role of *Ubqln1* in BCR-mediated apoptosis, cells were stained with Annexin V and 7AAD at multiple timepoints following addition of anti-IgM. For both WT and Ubqln1⁻/⁻ cells, we observed peak Annexin V staining at 8–16 hr post-stimulation, and at each time point tested there were fewer *Ubqln1*⁻/⁻ apoptotic cells (*Figure 5A*). Moreover, *Ubqln1*⁻/⁻ B cells were significantly less likely to enter the cell cycle, as measured by DNA content after BCR ligation, indicating a block in G0 or G1

**Table 2.** GO-term CC cluster analysis of proteins increased by *Ubqln1* knockout in murine B cell lysate.

DAVID pathway analysis was used on a dataset of proteins enriched at least 1.25-fold with p value < 0.05 (33 proteins) in anti-IgM stimulated, *Ubqln1*$^{-/-}$ murine B-cell lysate, using the background of all identified peptides from the MS run (approx. 8000 proteins). GO-term Cellular Compartments were used for a cluster analysis; clusters with Enrichment Scores of > 1.3 (p=0.05 equivalent) were listed with their common term.

| Term | Enrichment Score |
| --- | --- |
| Endoplasmic Reticulum | 1.63 |
| Mitochondrial Membrane | 1.33 |

DOI: https://doi.org/10.7554/eLife.26435.015

(*Figure 5B*). These data demonstrate that *Ubqln1*-deficient B cells were less likely than wild-type cells both to enter cell cycle and to apoptose in response to BCR ligation.

BCR crosslinking controls survival, apoptosis, proliferation, and antibody production via multiple signal transduction pathways, including the MAPK, NF$_k$B, and PI3K pathways (*DeFranco, 1997*). SYK, ERK1/2, AKT, and NF$_k$B activation showed similar kinetics in WT and *Ubqln1*$^{-/-}$ B cells

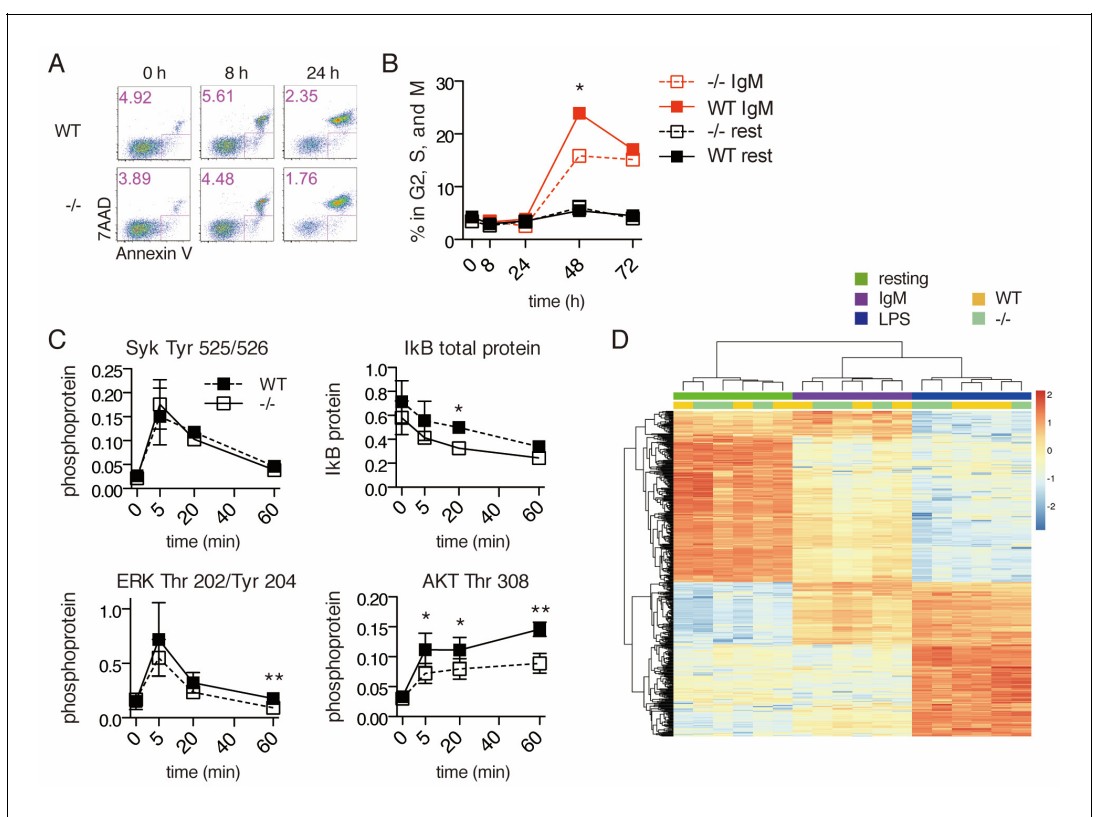

**Figure 5.** Signal transduction following BCR stimulation is normal in *Ubqln1*$^{-/-}$ cells. (**A**) Apoptosis assay using Annexin V and 7AAD staining at 8 and 24 hr following BCR stimulation. Shown are one representative dot plot each from triplicate wells, from one representative experiment of three. Apoptotic cells are gated in pink and quantified in the top left corner of each dot plot. (**B**) Cell cycle assay using DNA content. One representative of three experiments is shown with mean ± SEM of live cells containing > 1 n DNA as measured by Ruby Dyecycle signal. (**C**) Kinase activation following BCR stimulation. Anti-IgM was added to WT and *Ubqln1*$^{-/-}$ B cells for the indicated times, and phospho-SYK, total I$_k$B, phospho-ERK, and phospho-AKT were quantified by western blot. Graphs represent mean ± SEM of two independent experiments for SYK, and four experiments for I$_k$B, ERK, and AKT. (**D**) RNASeq was performed on RNA isolated from B-cells 4 hr after incubation with buffer, 10 µg/mL anti-IgM, or 20 µg/mL LPS. After estimation of mRNA levels, unsupervised clustering was performed according to Materials and methods. Significance was determined using paired T tests.

DOI: https://doi.org/10.7554/eLife.26435.016

(*Figure 5C*). To determine whether the mild abnormalities seen in the extent of AKT and NF$_k$B activation were significant, an RNAseq experiment was performed on WT and *Ubqln1*-deficient B cells at 4 hr post-stimulation with IgM and LPS. While marked differences were seen in gene expression when comparing resting, TLR4-stimulated, and BCR-stimulated cells, no consistent *Ubqln1*-dependent expression differences were found in resting cells or in response to either BCR or TLR stimulation (*Figure 5D*). These data demonstrate that proximal BCR signaling is intact in *Ubqln1*-deficient B cells and that a defect in BCR signaling cannot explain the proliferation and apoptosis defects caused by deletion of *Ubqln1*.

## *Ubqln1* is required for cyclin D accumulation downstream of BCR crosslinking

To determine the mechanism of G1 cell cycle blockade in *Ubqln1*$^{-/-}$ mice more precisely, we examined the induction of the central transcriptional regulator *c-Myc*, as well as the cell cycle regulator Cyclin D, both of which are required for BJAB proliferation (*Pfeifer et al., 2013*; *Schmitz et al., 2012*). Of the D-type cyclins, only *Ccnd2* is required for murine B1a cell development (*Solvason et al., 2000*; *Geng et al., 2003*; *Glassford et al., 2003*), implying a specialized role in BCR-induced proliferation. mRNA for both *Ccnd2* and *Myc* was induced to comparable levels in *Ubqln1*$^{-/-}$ and WT cells (*Figure 6A–B*). c-MYC protein also accumulated normally (*Figure 6C–D*). In striking contrast, CCND2 protein levels did not increase in *Ubqln1*$^{-/-}$ B cells after BCR signaling. (*Figure 6C–D*). CCND3, which is also expressed in murine B cells and is required for BJAB proliferation (*Schmitz et al., 2012*), was also decreased in *Ubqln1*$^{-/-}$ B cells (*Figure 6D*). Thus, *Ubqln1* is required for CCND2 and CCND3 protein upregulation in response to BCR signaling, even though the mRNA response is *Ubqln1*-independent. This abnormality likely underlies the defective cell cycle entry and the in vivo B cell phenotype, as CCND2 is known to be required both for the B cell G1/G2 transition in response to BCR ligation and for the development of B1a cells (*Solvason et al., 2000*).

Since CCND2 has no transmembrane domains and is a soluble cytosolic protein, we considered whether the accumulation of hydrophobic mitochondrial proteins might lead to enhanced degradation of CCND2. To test whether the lower expression of CCND2 protein in *Ubqln1*$^{-/-}$ cells after BCR stimulation was due to enhanced proteasomal degradation of CCND2, B cells were incubated with MG132, a proteasome inhibitor, after BCR stimulation. When MG132 was added 4 hr after cell activation, c-MYC protein level was increased for both WT and *Ubqln1*$^{-/-}$ B cells, but there was no significant effect on CCND2 in either WT or *Ubqln1*$^{-/-}$ B cells (*Figure 7A–B*). To test whether the proteasome has a more significant role when cellular CCND2 levels were higher, MG132 was added for the final 4 or 8 hr of a 24 hr incubation with anti-IgM. MG132 incubation for 4 or 8 hr at this later timepoint dramatically increased c-MYC protein levels in WT cells (*Figure 7C–D*), indicating that proteasomal degradation is a key contributor to low c-MYC levels at late timepoints. *Ubqln1*$^{-/-}$ c-MYC levels were not significantly increased by MG132 incubation at this late timepoint (*Figure 7D*), suggesting that the failure to increase protein concentration in the absence of *Ubqln1* at this timepoint was not entirely due to proteasomal degradation. Proteasome inhibition did not increase CCND2 levels to the same extent as those of c-MYC in either WT or *Ubqln1*$^{-/-}$ B cells, indicating that little CCND2 is degraded during this portion of the cell cycle (*Figure 7D*). Furthermore, MG132 incubation in *Ubqln1*$^{-/-}$ cells did not rescue CCND2 protein to WT levels (*Figure 7D*). Thus, increased proteasomal degradation did not account for the lack of CCND2 in *Ubqln1*$^{-/-}$ B cells after BCR stimulation.

## *Ubqln1* loss is associated with a defect in protein synthesis after BCR ligation

To determine whether the defects in CCND2 and D3 reflected a general problem with translation, incorporation of the methionine analog azido-homo-alanine (AHA) was assessed in B cells at 20 hr after the addition of anti-IgM. Although there was no difference in cell viability between WT and *Ubqln1*$^{-/-}$ under these conditions (*Figure 8A*), nor any defect in protein synthesis in resting *Ubqln1*$^{-/-}$ B cells, there was a marked defect in global protein synthesis after BCR stimulation in the absence of *Ubqln1* (*Figure 8B–D*). New protein production was modestly Ubqln1-dependent upon LPS stimulation (*Figure 8E*) but was still elevated about fourfold compared to BCR-stimulated cells,

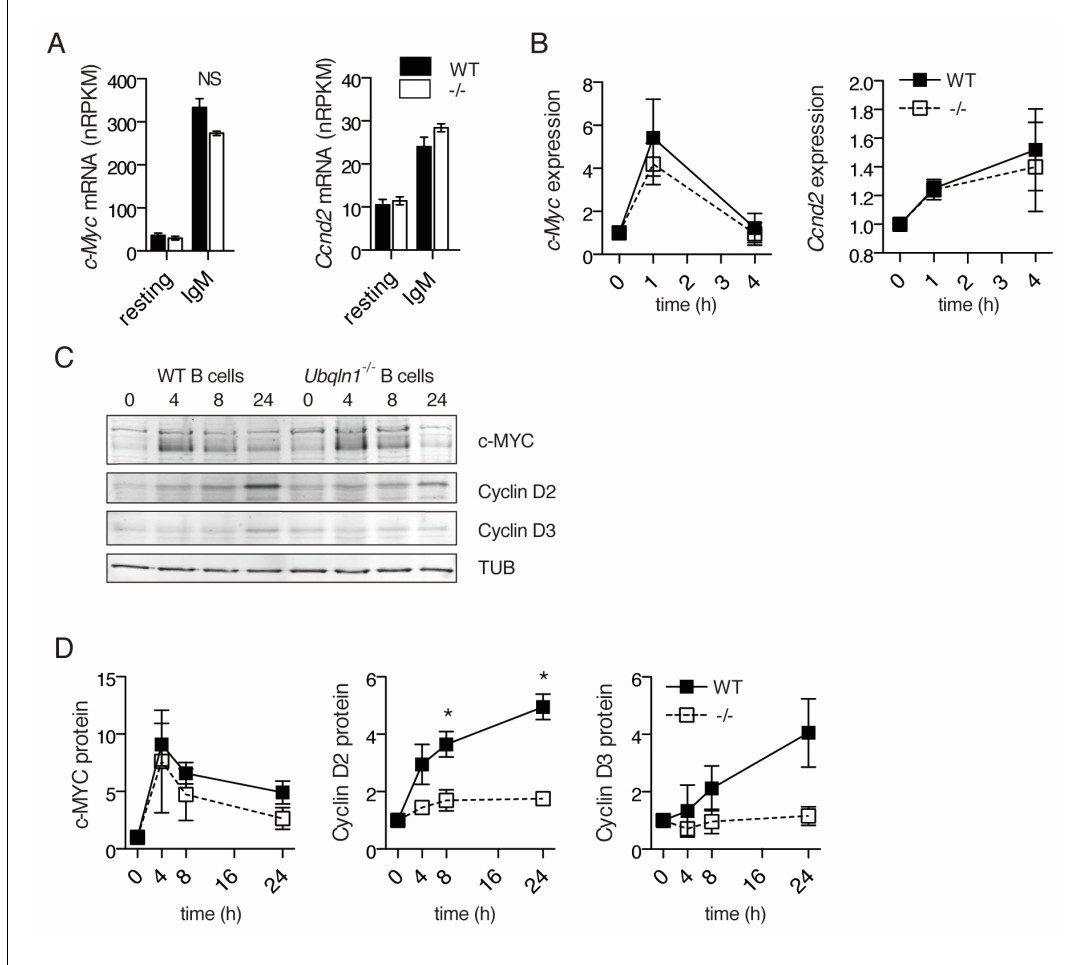

**Figure 6.** *Ubqln1⁻/⁻* B cells fail to accumulate Cyclin D proteins in response to BCR stimulation. (**A**) Expression of *c-Myc* and *Ccnd2* mRNA determined by RNASeq performed as in (**Figure 5**) (N = 3 independent experiments). (**B**) mRNA expression of *c-Myc* and *Ccnd2* determined by QPCR at multiple timepoints following BCR stimulation. Shown is mean ± SEM of three independent experiments. (**C**) Representative western blot showing timecourse of c-MYC and Cyclin D2 and D3 protein accumulation following BCR crosslinking. Tubulin is used as a loading control. (**D**) Quantification of protein accumulation as a function of time for c-MYC and Cyclin D2. Shown is mean ± SEM of four independent experiments. For (**A–D**) significance was determined by a paired Student's T test.

DOI: https://doi.org/10.7554/eLife.26435.017

indicating that *Ubqln1⁻/⁻* B cells are intrinsically capable of elevating their protein synthesis rate, and translation is particularly inhibited upon BCR stimulation.

To confirm that our failure to observe AHA-labeled proteins was due to a defect in translation and not due to increased proteasomal degradation of nascent protein, we tested incorporation of AHA in the presence of the proteasome inhibitor MG132. Proteasome inhibition actually decreased accumulation of AHA-labeled proteins over a 4-hr period after BCR ligation in both WT and *Ubqln1⁻/⁻* cells (**Figure 8F**), as has been reported (**Ding et al., 2006**; **Vabulas and Hartl, 2005**; **Wu et al., 2009**). MG132 did not diminish the difference in accumulation of AHA-labeled proteins between WT and *Ubqln1⁻/⁻*, indicating that this difference in new protein accumulation was not due to increased proteasomal degradation in the absence of *Ubqln1*. We confirmed the defect in protein synthesis in the absence of UBQLN1 in BJAB cells by performing the same metabolic labeling experiment and found that BJABs depleted of UBQLN1 also synthesize less new protein (**Figure 8G**).

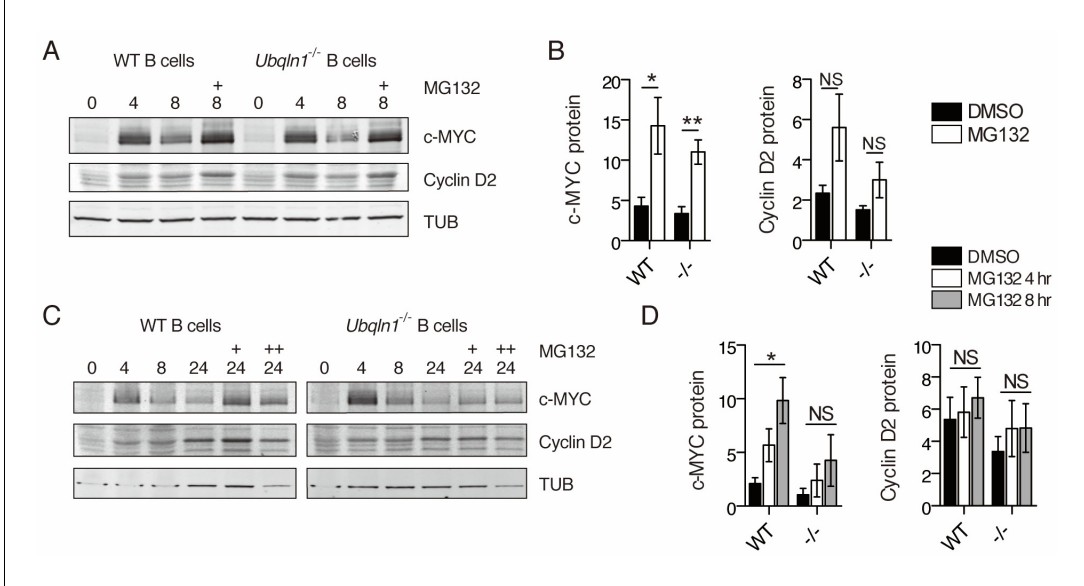

**Figure 7.** Proteasome inhibition does not rescue Cyclin D2 protein levels in *Ubqln1*$^{-/-}$ cells. (A–B) Anti-IgM stimulated cells were incubated with 5 μM MG132 between hours 4 and 8 post-stimulation. (C–D) Cells were incubated with MG132 as in (A–B) between 20 and 24 hr post stimulation (C, single + and D, open bars) or between 16 and 24 hr post-stimulation (C, double ++, and D, gray bars). Bar graphs in (B, D) are summaries of three to five independent experiments with mean ± SEM. Statistics were determined by paired T test.

DOI: https://doi.org/10.7554/eLife.26435.018

## Discussion

After their discovery in the 1990s, Ubqlns and their yeast orthologue, Dsk2, were implicated in proteasomal and autophagosomal degradation of multiple proteins (*Elsasser et al., 2002*; *Elsasser et al., 2004*; *Shi et al., 2016*; *Funakoshi et al., 2002*; *Itakura et al., 2016*; *Hjerpe et al., 2016*; *Rothenberg et al., 2010*; *Mah et al., 2000*). More recently, Ubqlns have been found to target mislocalized mitochondrial membrane proteins to the proteasome to be degraded (*Itakura et al., 2016*). Presumably, this protects cells against the accumulation of hydrophobic proteins in the cytosol, where they could aggregate, bind non-specifically to membranes, or induce other cellular damage. Indeed, this role for Ubqlns may not be limited to mitochondrial membrane proteins, as UBQLN4 has recently been reported to have a general role in degradation of mislocalized tail-anchored proteins (*Suzuki and Kawahara, 2016*) and UBQLN2 in the degradation of aggregation-prone proteins (*Hjerpe et al., 2016*). To discover how these potential roles for Ubqlns affected normal cell functions in vivo, we generated a cell line and mice deficient in *Ubqln1*, the most broadly and highly expressed member of the Ubqln family. The loss of *UBQLN1* from the BJAB B-cell lymphoma led to a rapid but reversible loss of proliferative capacity, and a comprehensive quantitative proteomic analysis revealed that the major effect of *UBQLN1* was to suppress the accumulation of hydrophobic mitochondrial proteins in the cytosol. This conclusion is consistent with the work of Itakura et al. (*Itakura et al., 2016*), who reported an accumulation of the mislocalized mitochondrial protein ATP5G1 upon depletion of all Ubqlns (UBQLN1, UBQLN2, and UBQLN4). While there appeared to be significant downregulation of UBQLN2 in this experiment, this might result from the different subcellular localization of UBQLN1 and UBQLN2. UBQLN2 is reported to localize mostly to the nucleus (*Hjerpe et al., 2016*), and since Ubqlns are capable of dimerization (*Ford and Monteiro, 2006*), the depletion of UBQLN1 could lead to destabilization of the small cytosolic UBQLN2 pool, leading to the apparent difference in UBQLN2 loss from *Figure 1C* (which measures only cytosolic UBQLN2) to *Figure 4A* (which measures total cellular UBQLN2).

A major finding from BJAB cells was the profound effect of UBQLN1 depletion on proliferative capacity, which may reflect that BJAB cells are classified as an 'Oxphos' DLBCL subtype (*Brien et al., 2007*), characterized by high expression of nuclear-encoded mitochondrial genes (*Monti et al., 2005*). Thus, it is possible that BJABs are hypersensitive to the loss of UBQLN1 due to

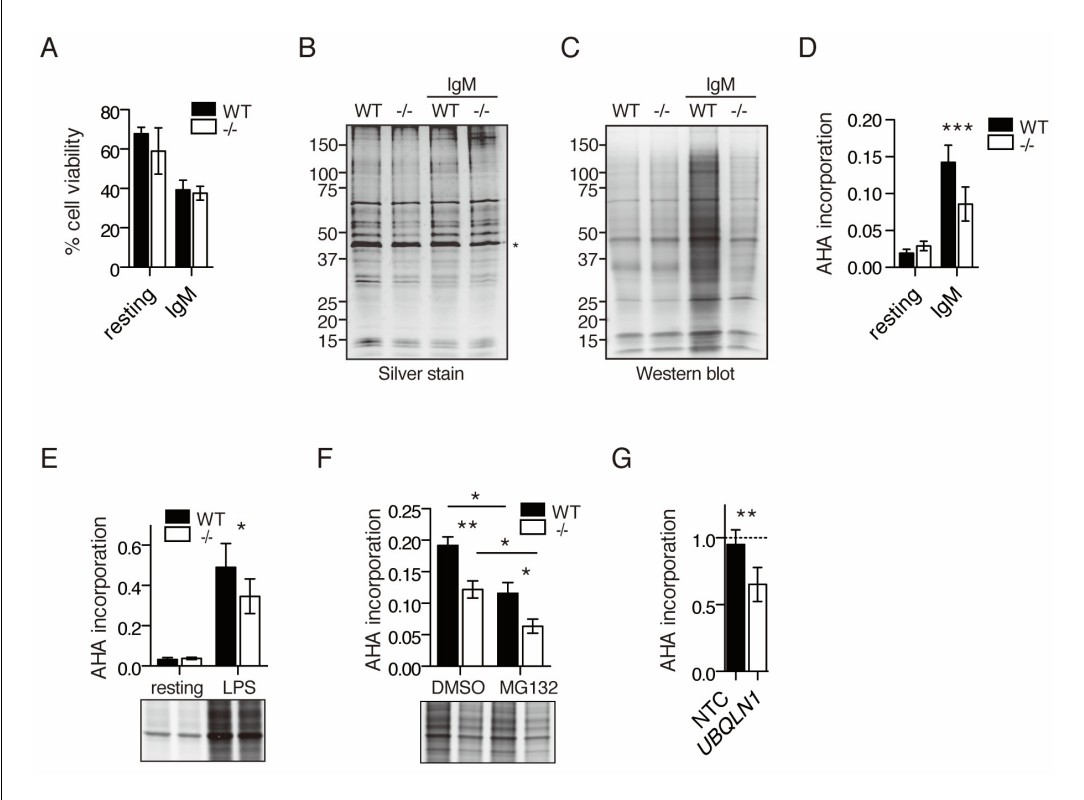

**Figure 8.** Decreased protein synthesis following BCR stimulation in *Ubqln1⁻ᐟ⁻* B cells. (A) Viability of cells in AHA translation assay. Stimulated (IgM) or unstimulated (resting) cells were starved of methionine for 45 min at 19 hr post-stimulation, then incubated with 2 μM AHA for 4 hr. Cell viability was determined by exclusion of trypan blue after 4 hr of incubation with AHA. Shown is mean ± SEM of three independent experiments. (B–C) After incubation with AHA, click chemistry was performed on cell lysates to add biotin-alkyne to incorporated AHA, and lysates were separated by SDS-PAGE. (B) silver stain; (C) western blot with streptavidin from one representative experiment. * denotes the Actin band used as a loading control for quantification of total protein and AHA incorporation. (D) Quantification of AHA incorporation. Shown is mean ± SEM for three to five independent experiments. (E) Quantification of AHA incorporation in resting and LPS-stimulated WT and *Ubqln1⁻ᐟ⁻* cells. Shown is mean ± SEM from four independent experiments with a representative western blot at bottom, as in (C). (F) Effect of MG132 on AHA incorporation. Cells were incubated with 5 μM MG132 during methionine starvation and AHA incorporation. Shown is mean ± SEM for four independent experiments with a representative western blot shown. Significance was determined with paired T test. (G) BJABs were incubated with doxycycline for 48 hr, live cells separated, and newly synthesized protein labeled with AHA. Shown is mean ± SEM of three experiments. Significance was determined via paired T-test.
DOI: https://doi.org/10.7554/eLife.26435.019

high expression levels of mitochondrial proteins. BJAB cells may also be exceptionally dependent on UBQLN1, as opposed to other members of the Ubqln family, for degradation of mislocalized mitochondrial proteins. To understand this better, we studied primary B cells from *Ubqln1* deficient mice. In these mice, we found a defect only in B1a B cells; other B cell compartments were normal. B1a cells are characterized by a unique reliance on BCR stimulation for their self-renewal in the peritoneum, which strongly suggested that *Ubqln1* function was tied to BCR signaling-dependent proliferation. This conclusion was supported by the dependence of BJAB proliferation on constitutive tonic BCR signaling (*Davis et al., 2010*). Indeed, BCR ligation in primary *Ubqln1*-deficient B cells led to a defect in cell cycle entry that comprehensive unbiased proteomic analysis showed was associated with accumulation of hydrophobic and mitochondrial proteins, as in BJAB.

The role of *Ubqln1* in degradation of mislocalized mitochondrial protein is particularly critical in murine B cells when BCR ligation initiates proliferation, as protein synthesis and proliferation in response to LPS were normal. A likely explanation for this specificity is that BCR ligation induces much greater and more prolonged mitochondrial depolarization than other mitogenic stimuli, such as TLR4 ligation. Since mitochondrial depolarization attenuates the import of mitochondrial proteins (*Schleyer et al., 1982*), it is the specific accumulation of mislocalized protein that reveals a role for

UBQLN1 in vivo. Both BCR and TLR4 stimulation increase mitochondrial mass and the production of nuclear-encoded mitochondrial proteins, but the mitochondrial depolarization induced by BCR signaling compromises import of nascent protein, which increases the importance of degradation of mislocalized proteins in order to maintain cellular homeostasis. The increased requirement for *Ubqln1* in BCR-stimulated B cells therefore appears to be dependent on the combination of increased mitochondrial protein synthesis and decreased mitochondrial protein import. Increased mitochondrial protein production alone is not sufficient to reveal a requirement for *Ubqln1* in B cell proliferation, as LPS-stimulated B cells proliferate normally. Stimulation of B cells with IL-4 (53,64) or CD40 (*Eeva et al., 2003*) in addition to anti-IgM ameliorates BCR signaling-induced mitochondrial depolarization, which could explain why the addition of these co-stimuli to anti-IgM rescued B-cell proliferation in the absence of *Ubqln1*. It is interesting that the mitochondria themselves show no obvious functional defect in the absence of Ubqln1 in BCR-stimulated cells. This suggests that under these conditions, the import machinery is capable of meeting the demand of increased mitochondrial biogenesis, and that nuclear-encoded mitochondrial proteins are made in excess to the mitochondrial requirements. In any case, the finding of normal mitochondrial function in *Ubqln1*-deficient cells emphasizes that the toxic event is not intrinsic to mitochondria but related to the cytosolic accumulation of mislocalized mitochondrial proteins.

How does the accumulation of mislocalized proteins lead to a defect in protein synthesis, cell cycle entry, and subsequent proliferation? The kinetics of mRNA induction following BCR stimulation were normal in the absence of Ubqln1, which suggests that UPR and other stress response pathways acting through effects on gene expression programs are unlikely to be effectors of this response. Instead, our results indicated that the important *Ubqln1*-dependent step in proliferation in response to BCR signaling was translation of at least some of the induced mRNAs, including Cyclin Ds, which are essential for progression out of G1. Accumulation of mislocalized mitochondrial proteins can depress cytosolic protein synthesis through initiation of a poorly understood cellular stress response (*Wrobel et al., 2015*; *Wang and Chen, 2015*; *Topf et al., 2016*). These studies demonstrated that inhibition of mitochondrial protein import in yeast leads to a general inhibition of cytosolic protein synthesis as measured by metabolic labeling with $S^{35}$ methionine. Wang et al. also quantified by mass spectrometry the proteome of yeast expressing a mutant ADP/ATP exchanger that collapses mitochondrial polarity and found a specific accumulation of nucleus-encoded mitochondrial proteins, two of which had MTS sequences still attached (*Wang and Chen, 2015*). These studies support the hypothesis that the protein synthesis defect observed upon BCR stimulation in the absence of UBQLN1 is due to the accumulation of mislocalized mitochondrial protein and suggest that this is a more general phenomenon. B cells likely are exceptionally dependent on UBQLN1, as opposed to other members of the Ubqln family, for degradation of mislocalized mitochondrial proteins because of the high relative expression of Ubqln1 in these cells. We also considered the possibility that ER stress, inhibition of mTOR activity, or the loss of energy levels might contribute to the observed general inhibition of protein synthesis, but found no Ubqln1-dependent changes in EIF2α phosphorylation, UPR gene expression, mTOR pathway phosphorylation, and only slight changes in cellular ATP levels.

Further studies will be needed to determine the full extent of physiological stresses that reveal a requirement for Ubqln function. Mitochondrial depolarization downstream of calcium signaling, or calcium overload, have been implicated in receptor-mediated excitotoxicity (*Stanika et al., 2009*; *White and Reynolds, 1996*) and hyperexcitability (*Carriedo et al., 2000*; *Fuchs et al., 2013*), which are hallmarks of neurological diseases ranging from ischemic stroke to ALS (*Doble, 1999*). Also, it appears that defects in mitophagy, an autophagic degradation of mitochondria that is triggered by mitochondrial depolarization, underlie familial Parkinson's disease (*Narendra et al., 2012*; *Hauser et al., 2013*). As in BCR-stimulated B cells, neurons stimulated with mitochondrial-depolarizing agents may require Ubqln expression to ameliorate the buildup of mitochondrial proteins in the cytosol. It is noteworthy that loss-of-function mutations in *UBQLN2*, which is highly expressed in neurons and muscle (*Marín, 2014*), appear to play an etiological role in some forms of familial ALS-FTD (*Deng et al., 2011*). Further tests are warranted to explore the cytosolic accumulation of mitochondrial proteins in neurons deficient of UBQLN2 function, and whether mitochondrial depolarization exacerbates this accumulation. In conclusion, we hypothesize that prolonged mitochondrial depolarization, paired with amplified expression of nuclear-encoded mitochondrial proteins, may represent a unifying stress that makes Ubqln expression essential for the continued health of the cell.

## Materials and methods

### Antibodies and reagents

Anti-CD19 RRID:AB_2629813 (FITC), anti-IgM RRID:AB_2075943 (Percp/Cy5.5), anti-CD21 RRID:AB_1953277 (PE/Cy7), anti-CD23 RRID:AB_2103307 (Pacific Blue), anti-CD5 RRID:AB_2563928 (APC), anti-CD138 RRID:AB_10960141 (APC), and anti-B220 RRID:AB_313007 (APC/Cy7) antibodies for flow cytometry were purchased from Biolegend (San Diego, CA). Western blot antibodies against c-MYC RRID:AB_1903938, Cyclin D2 RRID:AB_10830736, Cyclin D3 RRID:AB_10841292, phospho-SYK (Tyr515/516) RRID:AB_2197222, I$_k$B RRID:AB_823540, phospho-AKT (Thr308) RRID:AB_2315049 and phospho-ERK1/2 (Thr202/Tyr204) RRID:AB_2315112 were purchased from CST (Danvers, MA). Rabbit polyclonal UBQLN1 antibody was generated at Genentech against recombinant human UBQLN1. Mouse anti-tubulin RRID:AB_11204167 and polyclonal rabbit VDAC RRID:AB_2304154 antibodies were purchased from Thermo Fisher Scientific (Waltham, MA). ATP5G1 mouse antibody RRID:AB_1839997 was purchased from Sigma Aldrich (St. Louis, MO). IgD-biotin antibody RRID:AB_2563343 was purchased from Biolegend. UBQLN2 antibody RRID:AB_670669 was purchased from Novus Biologicals (Littleton, CO) and UBQLN4 antibody RRID:AB_11174448 was purchased from Genetex (Irvine, CA). Secondary anti-mouse DyLight 680 RRID:AB_11183140 and anti-rabbit DyLight 800 RRID:AB_1660964 antibodies for western blot were purchased from Rockland Immunochemicals (Limerick, PA). DAPI was purchased from eBioscience (Thermo Fisher). 6- and 10-plex TMT reagents were purchased from Thermo Fisher.

### Cell lines

All cell lines used in the study were banked at Genentech and tested for contamination by STR and SNP analysis (*Yu et al., 2015*), as well as for mycoplasma contamination. BJAB cells RRID:CVCL_5711 (originally from DSMZ) were confirmed as mycoplasma-negative and cultured in complete medium with RPMI, Pen/strep, L-glutamine (Gibco, Thermo Fisher), and 10% FBS (Sigma). HeLa cells RRID:CVCL_0030 (originally from ATCC, Manassus, VA) were also confirmed as mycoplasma-negative and were cultured in complete medium with DMEM, Pen/strep, L-glutamine, and 10% FBS. To generate stably expressing *UBQLN1* or control shRNA BJAB lines, cells were infected with lentivirus containing pINDUCER10 (*Meerbrey et al., 2011*) with either a non-targeting construct (NTC) or an shRNA construct against *UBQLN1*. Six *UBQLN1* shRNA constructs were designed using DSIR (*Vert et al., 2006*) and checking for off-target effects by comparing with TargetScan Human custom (*Lewis et al., 2005*) and a nucleotide BLAST (*Altschul et al., 1990*). The six constructs were tested for their ability to knockdown UBQLN1 expression with doxycycline (Takara Bio USA, Mountain View, CA) and the most effective shRNA construct was selected for further study. Proliferation of BJAB and HeLa cells was determined with CellTiter Glo (Promega, Madison WI) in duplicate wells.

### Mice

A cassette containing floxed exon 2 of the murine *Ubqln1* gene was inserted into C57BL/6 ES cells by homologous recombination and selected with neomycin resistance. Following selection, neo resistance was removed via FRT recombination and exon two deleted via transient Cre expression. ES cells lacking exon 2 of *Ubqln1* were then used to generate knockout mice. *Ubqln2*$^{-/-}$ and *Ubqln4*$^{-/-}$ mouse models were generated in a fashion similar to *Ubqln1*$^{-/-}$ mice by floxing exon 1 of *Ubqln2* and exons 2 and 3 of *Ubqln4* in the C57BL/6 backgrounds. Mice were genotyped using small tail clippings and the Sigma Extract-N-Amp tissue PCR kit. *Ubqln1*$^{-/-}$ mice were housed according to IACUC guidelines and all experiments were performed according to IACUC standards and under protocol numbers TH16-0169, 15–2041, 15–3110, 16–1851, and 16–2557. For tissue collection, mice were euthanized with $CO_2$ according to IACUC guidelines, followed by cervical dislocation. All mice were at least 8 weeks of age for experiments.

### Flow cytometry

For splenic, LN, and peritoneal B-cell identification and relative quantification, cells were isolated from spleen, LN, or peritoneal lavage and stained in FACS buffer at recommended concentrations for 15 min. For apoptosis assay, cells were stained with Annexin V-APC (BD Biosciences, San Jose CA) according to manufacturer's staining protocol, and counterstained with 7AAD (Biolegend). For

cell cycle analysis, cells were harvested into complete medium with 1:20 7AAD and 1:500 Vybrant DyeCycle Ruby (Thermo Fisher) and incubated for at least 30 min at 37°C in the dark. Cells were analyzed using an LSRII on low or medium speed and by first gating on live, singlet cells. A minimum of 10,000 events was recorded for cellular assays and 500,000 for immunophenotyping.

## ELISA

Blood for serum samples was collected from tail vein or cardiac puncture into serum collection tubes. Serum was frozen at −80 until use. For total IgG and IgM, a standardized kit and protocol from Bethyl Laboratories (Montgomery, TX) was used.

## Primary B-cell stimulation

B cells were isolated from spleen of at least 8-week-old mice using the Miltenyi Biotec murine B-cell isolation kit. Following isolation, cells were counted and either immediately plated in triplicate wells in complete B-cell medium (DMEM:H12 with 10% Hyclone FBS, L-glutamine, penicillin/streptomycin, 12 mM HEPES, and 55 µM Beta-mercaptoethanol) at $2 \times 10^6$ cells/mL, or stained with CFSE (Thermo Fisher) before plating. B cells were stimulated with either 5 or 10 µg/mL of goat anti-mouse IgM F (ab)2 (Sigma), 2 or 20 µg/mL K12 LPS (Invivogen, San Diego CA), 100 or 500 ng/mL recombinant human BAFF (Peprotech, Rocky Hill NJ), and were sometimes supplemented with 20 ng/mL of murine IL-4 (Peprotech) or 1 µg/mL LEAF-purified anti-mouse CD40 antibody RRID:AB_312942(Biolegend). For MG132 experiments, cells were incubated with either Hybrimax DMSO (Sigma Aldrich) or 5 µM of MG132 (EMD Millipore, Billerica MA).

## RNASeq

For RNASeq, cells were stimulated in triplicate for 4 hr with either 10 µg/mL anti-IgM F(ab)$_2$, 20 µg/mL LPS, or no stimulation. Triplicate samples were pooled for RNA purification. RNA purification was performed using the RNEasy kit with on-column DNAse digestion (Qiagen, Germantown MD). 0.5 µg of total RNA was used as an input material for library preparation using TruSeq RNA Sample Preparation Kit v2 (Illumina, San Diego CA), which was quality controlled with Fragment Analyzer and a Library quantification kit (KAPA Biosystems, Wilmington MA). The libraries were multiplexed and then sequenced on Illumina HiSeq2500 to generate 30M of single end 50 base pair reads. Sequencing reads were mapped to the reference mouse genome (GRCm38), using the GSNAP short read aligner (*Wu and Nacu, 2010*). Expression was measured in reads per kilobase per million total reads (RPKM). Analysis of processed RNAseq data (normalized and variance-stabilized by voom) was performed using R (v3.2.2) and Bioconductor (v3.2). Differential expression filtering was performed with the limma package using single-term linear models and cut-off thresholds of absolute log fold change greater than two and adjusted (Benjamini-Hochberg) p-value less than 0.001. Clustering was performed on data for these sets of differentially expressed genes by complete agglomerative iteration on Euclidean distances of log-transformed, scaled, and centered data.

## QPCR

For QPCR, cells were stimulated and mRNA was isolated as for RNASeq. cDNA was generated using the ABI high-capacity cDNA reverse transcription kit. 10 ng of cDNA was used as input for Taqman gene expression assays (ABI, now Thermo Fisher) run in triplicate with internal *Gapdh* controls using standard cycling parameters on an ABI 7500 real-time PCR system. Raw triplicate Ct values were analyzed and normalized against *Gapdh*. Taqman primer/probe sets were as follows: *Gapdh* (VIC): Mm99999915_g1, *c-Myc* (FAM): Mm00487804_m1, *Ccnd2* (FAM): Mm00438070_m1, *Ubqln1*: Mm00455863_m1, *Ubqln2*: Mm00834570_s1, *Ubqln4*: Mm00473805_m1.

## Western blotting

Primary B cells and BJAB cells were plated at $2 \times 10^6$ cells/mL density in 2 mL volumes of 12-well plates. Wells were harvested, washed, and reconstituted in 2x sample buffer with β-mercaptoethanol and boiled for 7 min, then snap frozen in dry ice. Each well of primary B cells was run as one lane of a 4–20% Novex Tris-glycine gel (Thermo Fisher). Proteins were transferred onto nitrocellulose membrane using the iBlot system (Thermo Fisher). Membranes were blocked for 30 min with Rockland buffer and incubated with primary antibody overnight at 4°C. After washing, membranes were

stained in secondary antibody for 1 hr at room temperature. After washing again, membranes were visualized and quantitated using LICOR Odyssey (Lincoln, NE).

## B cell electron microscopy

For mitochondrial TEM images, resting or IgM-stimulated cells at 24 hr post-stimulation were washed in PBS and then fixed in 1/2 Karnovsky's fixative (2% paraformaldehyde, 2.5% glutaraldehyde in 0.1 M sodium cacodylate buffer, pH 7.2). The samples were then post-fixed in 1% reduced osmium tetroxide, stained with 0.5% uranyl acetate and then dehydrated through a series of ethanol steps (50, 70, 90% and 100%) followed by two rinses with propylene oxide. Cells were embedded in Eponate 12 (Ted Pella, Redding CA) and curing of the samples was at 65°C. Semithin (500 nm) and ultrathin (80 nm) sections were obtained with an Ultracut microtome (Leica Biosystems, Buffalo Grove IL). The semithin sections were stained with Toluidine Blue and examined by bright field microscopy to obtain an overview of the cells to be examined by TEM. Ultrathin sections on grids were counterstained with 0.2% lead citrate and examined in a JEOL JEM-1400 transmission electron microscope (TEM) at 80 kV. Digital images were captured with a GATAN Ultrascan 1000 CCD camera.

## Seahorse mitochondrial stress test

Seahorse XF24 (Agilent Technologies, Santa Clara CA) plates were coated for 1 hr with 3.5 µg/cm$^2$ CellTak (Corning, Corning NY), washed with water and air-dried before cells were adhered. To adhere cells, murine B cells were harvested 24 hr after rest or stimulation with anti-IgM and resuspended in warm RPMI. Cells were counted and resuspended at $5*10^6$ cells/mL RPMI, then 100 µL was plated in quintuplicate wells, which were randomized to prevent plating effects. After seeding cells, plates were incubated for 30 min at 37 degrees, then spun at 1300 rpm for 1 min. After this, RPMI was removed and cells were carefully washed twice with warm complete Seahorse medium (Agilent) supplemented with sodium pyruvate at 10 mM. At the same time, the Seahorse cassette which had been equilibrated overnight was loaded with Oligomycin, FCCP, and Antimycin/Rotenone mixture for final concentrations of 1 µM/well. Then, a standard mitochondrial stress test protocol was run at an XF24 station with the plate. While the plate ran, a small aliquot of the WT and Ubqln1$^{-/-}$ cells that were used for seeding plates was stained with DAPI and run on the flow cytometer to determine live cell number per seeding volume. After Seahorse results were generated, they were normalized to live cell #/well generated from FACS data.

## AHA-labeling of cell lysate

For murine primary B cell experiments, cells were isolated and either rested or stimulated for 16–20 hr with 10 µg/mL anti-IgM. For BJABs, cells incubated either in media or with 100 ng/mL doxycycline for 48 hr were centrifuged over a ficoll gradient (GE Healthcare Lifesciences, Marlborough MA) to generate a pure live cell population. Cells were then washed and starved of methionine for 45 min with methionine-free RPMI (Thermo Fisher) supplemented with methionine-free FBS (Thermo Fisher). 2.5 µM AHA (Thermo Fisher) was then added back to the cells for 4 hr. Cells were harvested and counted by trypan blue exclusion, then cell pellets were frozen at −80°C until use. Total protein was labeled using the Click-IT protein labeling kit (Thermo Fisher) after normalizing the protein concentrations of cell lysates using a micro-BCA protein quantitation kit (Thermo Fisher). After labeling with 200 µM biotin-alkyne reaction buffer, protein was precipitated with chloroform/methanol extraction and resuspended in 35 µL of 2x sample buffer with β-mercaptoethanol, then heated for 7 min at 95°C. 5 µL of this sample was used for Silver stain (ProteoSilver by Sigma Aldrich), which was quantified by a Bio-Rad (Hercules, CA) GelDoc XR, and 30 µL used for Western blot, which was detected with streptavidin-A700 at 1:10,000 concentration by LICOR Odyssey.

## Cellular fractionation

Two × 10$^7$ BJAB cells were harvested after 48 hr in the presence of 100 ng/mL doxycycline or normal media, and washed twice with sterile PBS. Cells were then fractionated according to the QProteome Mitochondrial isolation kit (Qiagen). For ATP5G1 quantitation, resulting protein samples were heated for 7 min at 95°C and western blotted. For mass spectrometry, proteins were precipitated and resuspended in urea buffer for digestion.

## Sample preparation for mass spectrometry analysis

Murine B cell samples for multiplexed quantitative mass spectrometry analysis were processed and analyzed through the Thermo Fisher Scientific Center for Multiplexed Proteomics at Harvard Medical School (*Weekes et al., 2014*). First, cells were harvested following IgM stimulation, washed twice with PBS, and snap-frozen at −80° C. Lysis buffer (8 M Urea, 1% SDS, 50 mM Tris pH 8.5, Protease and Phosphatase inhibitors from Roche) was added to the cell pellets to achieve a cell lysate with a protein concentration between 2–8 mg/mL. A BCA assay (Pierce, Thermo Fisher) was used to determine the final protein concentration in the cell lysate. Fractionated human BJAB cytosol samples were processed using the QProteome Mitochondrial isolation kit (Qiagen) and analyzed in the Finley and Gygi laboratories. Cytosolic proteins were precipated with methanol/chloroform and resuspended in urea buffer (8 M urea, 50 mM EPPS pH 8.0, 75 mM NaCl, protease and phosphatase inhibitors (Roche)). Then, all proteins in 8 M urea buffer were reduced and alkylated as previously described (*Weekes et al., 2014*) and were precipitated using methanol/chloroform. The precipitated protein was washed with one volume of ice cold methanol. The washed precipitated protein was allowed to air dry. Precipitated protein was resuspended in 4 M Urea, 50 mM Tris pH 8.5.

Proteins were first digested with LysC (1:50; enzyme:protein) for 12 hr at 25°C. The LysC digestion is diluted down to 1 M Urea, 50 mM Tris pH8.5 and then digested with trypsin (1:100; enzyme:protein) for another 8 hr at 37°C. Peptides were desalted using a $C_{18}$ solid phase extraction cartridges as previously described. Dried peptides were resuspended in 200 mM EPPS, pH 8.0. Peptide quantification was performed using the micro-BCA assay (Pierce). The same amount of peptide from each condition was labeled with tandem mass tag (TMT) reagent (1:4; peptide:TMT label) (Pierce). The TMT labeling reactions (either 10-plex or 6-plex) were performed for 2 hr at 25°C. Modification of tyrosine residue with TMT was reversed by the addition of 5% hydroxyl amine for 15 min at 25°C. The reaction was quenched with 0.5% TFA and samples were combined at a 1:1:1:1:1:1:1:1:1:1 ratio. Combined samples were desalted and offline fractionated into 24 fractions as previously described (*Paulo and Gygi, 2017*).

## Liquid chromatography-MS3 (LC-MS3)

Fractions from the basic reverse phase step (every other fraction) were analyzed with an LC-MS3 data collection strategy (*McAlister et al., 2014*) on an Orbitrap Fusion mass spectrometer (Thermo Fisher) equipped with a Proxeon Easy nLC 1000 for online sample handling and peptide separations. Approximately 5 μg of peptide resuspended in 5% formic acid +5% acetonitrile was loaded onto a 100 μm inner diameter fused-silica micro capillary with a needle tip pulled to an internal diameter less than 5 μm. The column was packed in-house to a length of 35 cm with a $C_{18}$ reverse phase resin (GP118 resin 1.8 μm, 120 Å, Sepax Technologies, Newark DE). For BJAB cytosol, peptides were separated on a 100 μm inner diameter microcapllary column packed with ~40 cm of Accucore150 resin (2.6 μm, 150 A, ThermoFisher Scientific, San Jose, CA). For murine lysate, the peptides were separated using a 180 min or 150 min linear gradient from 3% to 25% buffer B (100% ACN +0.125% formic acid) equilibrated with buffer A (3% ACN +0.125% formic acid) at a flow rate of 400 nL/min across the column. The scan sequence for the Fusion Orbitrap began with an MS1 spectrum (Orbitrap analysis, resolution 120,000, 400–1,400 m/z scan range with quadrapole isolation, AGC target $2 \times 10^5$, maximum injection time 100 ms, dynamic exclusion of 90 s). 'Top N' (the top 10 precursors) was selected for MS2 analysis, which consisted of CID ion trap analysis, AGC target $8 \times 10^3$, NCE 35, maximum injection time 150 ms), and quadrapole isolation of 0.5 Da for the MS1 scan. The top 10 fragment ion precursors from each MS2 scan were selected for MS3 analysis (synchronous precursor selection, SPS), in which precursors were fragmented by HCD prior to Orbitrap analysis (NCE 55, max AGC $1 \times 10^5$, maximum injection time 150 ms, MS2 ion trap isolation was set to 2 Da, resolution 60,000 or 15,000 for the TMT10-plex and 6-plex, respectively.

## TMT-SPS-MS3 data analysis

A suite of Sequest-based (*Eng et al., 1994*) software tools were used for .RAW file processing and controlling peptide and protein level false discovery rates, assembling proteins from peptides, and protein quantification from peptides as previously described (*Huttlin et al., 2010*). Each analysis used an SPS-MS3-based TMT method, which has been shown to reduce ion interference compared to MS2 quantification (*Paulo et al., 2016*). MS/MS spectra were searched against a Uniprot human

database (February 2014) with both the forward and reverse sequences. Database search criteria are as follows: tryptic with two missed cleavages, a precursor mass tolerance of 50 ppm, fragment ion mass tolerance of 1.0 Da, static alkylation of cysteine (57.02146 Da), static TMT labeling of lysine residues and N-termini of peptides (229.162932 Da), and variable oxidation of methionine (15.99491 Da). TMT reporter ion intensities were measured using a 0.003 Da window around the theoretical m/z for each reporter ion in the MS3 scan. Peptide spectral matches with poor-quality MS3 spectra were excluded from quantitation (<200 summed signal-to-noise across 10 channels and <0.5 precursor isolation specificity).

## In silico analyses

DAVID pathway cluster analysis of GO-term CC_all was performed on protein datasets from mass spectrometry results, with all filtered results from each experiment used as background (*Huang et al., 2009*). Mitochondrial protein prevalence was tested by comparing mass spectrometry results from each experiment against Mitocarta 2.0 of either mouse or human (*Calvo et al., 2016*). Hydropathy was determined in silico by determining the GRAVY score (*Kyte and Doolittle, 1982*) of each protein found to be significantly altered between tested conditions.

## Mitochondrial testing

Cells were stained with Mitotracker Deep Red (250 nM), Mitosox (5 µM) and JC-1 dye (10 µg/mL) (Thermo Fisher) according to manufacturer's instructions and resuspended in complete medium for flow cytometric analysis.

## Statistics

For all experiments, statistical significant was determined by using either a paired or unpaired Student's T test (specifics are detailed in Figure legends) with a cutoff of *$p<0.05$, **$p<0.01$, ***$p<0.001$.

## Accession numbers

The primary RNASeq data have been deposited in the Gene Expression Omnibus (GEO) database from NCBI at GSE86113. The mass spectrometry proteomics data have been deposited to the ProteomeXchange Consortium via the PRIDE (*Vizcaíno et al., 2016*) partner repository with the dataset identifier PXD007610.

## Acknowledgements

IP, AA, BH, MR, AK, MS, ZM, MR-G, DYL, DK, BM, and EB are employees of Genentech, a member of the Roche Group. AW is a former employee of Genentech. We would like to thank Tangsheng Yi, John Monroe, and Tim Behrens for helpful B cell discussions. We would like to thank Ryan Kunz and the Thermo Fisher Center for Multiplexed Proteomics for their assistance with TMT10-plex proteomics. We would like to thank Baris Bingol and Joy Tea for helpful discussions about mitochondria and assistance with Seahorse. We would also like to thank Yuxin Liang for assistance with lentivirus. JP is supported by an NIH/NIDDK grant K01 DK098285. DF is supported by a grant from the Harvard Brain Initiative.

## Additional information

### Competing interests

Alexandra M Whiteley: Previously employed by Genentech, Inc. Ivan Peng, Alexander R Abbas, Benjamin Haley, Mike Reichelt, Anand Katakam, Meredith Sagolla, Zora Modrusan, Dong Yun Lee, Merone Roose-Girma, Donald S Kirkpatrick, Brent S McKenzie, Eric J Brown: Employed by Genentech, Inc. The other authors declare that no competing interests exist.

## Funding

| Funder | Grant reference number | Author |
|---|---|---|
| Cancer Research Institute | | Alexandra M Whiteley |
| National Institute of Diabetes and Digestive and Kidney Diseases | K01 DK098285 | Joao A Paulo |
| Harvard Brain Initiative | | Daniel Finley |

The funders had no role in study design, data collection and interpretation, or the decision to submit the work for publication.

## Author contributions

Alexandra M Whiteley, Conceptualization, Formal analysis, Investigation, Methodology, Writing—original draft, Writing—review and editing; Miguel A Prado, Alexander R Abbas, Formal analysis, Methodology, Writing—review and editing; Ivan Peng, Joao A Paulo, Methodology; Benjamin Haley, Merone Roose-Girma, Conceptualization, Methodology, Writing—review and editing; Mike Reichelt, Anand Katakam, Visualization, Methodology; Meredith Sagolla, Conceptualization, Visualization, Methodology; Zora Modrusan, Brent S McKenzie, Conceptualization; Dong Yun Lee, Methodology, Writing—review and editing; Donald S Kirkpatrick, Conceptualization, Formal analysis, Methodology, Writing—review and editing; Steven P Gygi, Conceptualization, Formal analysis; Daniel Finley, Conceptualization, Formal analysis, Writing—original draft; Eric J Brown, Conceptualization, Formal analysis, Investigation, Writing—original draft

## Author ORCIDs

Alexandra M Whiteley (iD) http://orcid.org/0000-0002-4144-7605
Alexander R Abbas (iD) http://orcid.org/0000-0003-2600-810X
Eric J Brown (iD) http://orcid.org/0000-0003-2125-6286

## Ethics

Animal experimentation: All mouse studies were performed in accordance with IACUC guidelines, under protocols approved by the Genentech Institutional Animal Care and Use Committee. All experiments were performed according to IACUC standards and under approved Genentech protocol numbers TH16-0169, 15-2041, 15-3110, 16-1851, and 16-2557.

## Decision letter and Author response

Decision letter https://doi.org/10.7554/eLife.26435.025
Author response https://doi.org/10.7554/eLife.26435.026

# Additional files

## Supplementary files

• Transparent reporting form
DOI: https://doi.org/10.7554/eLife.26435.020

## Major datasets

The following datasets were generated:

| Author(s) | Year | Dataset title | Dataset URL | Database, license, and accessibility information |
|-----------|------|---------------|-------------|--------------------------------------------------|
| Calvo SE, Klauser CR, Mootha VK | 2015 | Mouse Mitocarta 2.0 | https://www.broadinstitute.org/files/shared/metabolism/mitocarta/mouse.mitocarta.2.0.html | Publicly available from the MitoCarta website (https://www.broadinstitute.org/scientific-community/science/programs/metabolic-disease-program/publications/mitocarta/mitocarta-in-0) |
| Calvo SE, Klauser CR, Mootha VK | 2015 | Human MitoCarta 2.0 | https://www.broadinstitute.org/files/shared/metabolism/mitocarta/human.mitocarta2.0.html | Publicly available from the MitoCarta website (https://www.broadinstitute.org/scientific-community/science/programs/metabolic-disease-program/publications/mitocarta/mitocarta-in-0) |

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
