## [Decision Letter]

Thank you for submitting your article "Ubiquilin1 promotes antigen-receptor mediated proliferation by eliminating mislocalized mitochondrial proteins" for consideration by *eLife*. Your article has been favorably evaluated by Ivan Dikic (Senior Editor) and three reviewers, one of whom is a member of our Board of Reviewing Editors. The reviewers have opted to remain anonymous.

The reviewers have discussed the reviews with one another and the Reviewing Editor has drafted this decision to help you prepare a revised submission.

Summary:

As a whole this manuscript has been well received by the reviewers, which further supports the initial decision of the main editor to send out for review. While this manuscript addresses interesting and important questions as well as also present a set of important data, the reviewers highlighted areas of improvement as detailed below. We believe that this data represents a reasonable request which is intended to improve the impact and the message of this manuscript. Aside from immunizations or antigenic challenges, which might take some time, all this data can be collected in a 2 months period.

General reviewers’ comments:

This study by Whiteley et al. has investigated the consequences of Ubqln1 knockdown and knockout in B cells and B cell derived cell lines. The primary finding is that acute knockdown of Ubqln1 in the BJAB cell line rapidly arrests growth, and this correlates with the accumulation in the cytosol of many hydrophobic proteins, many of which are intended for targeting to mitochondria. Subsequent studies in a subset of B cells isolated from Ubqln1 knockout mice verified this general phenomenon under conditions where BCR signaling was initiated. They find that BCR signaling, but not LPS stimulation, induces mitochondrial depolarization in addition to mitochondrial proliferation. The association of Ubiquilin1 with mitochondrial proteins has been previously established by Itakura et al., thus this aspect of the work represents a rather modest advance. The novelty comes from the analysis of Ubqln^-/-^ mice which are purported to have a B cell restricted phenotype revealed by a diminution in B1 cells and impaired responsiveness to BCR-induced proliferation.

In general the topic is clearly of significant interest because this family of proteins is poorly understood and implicated in diseases via unknown mechanisms. The recent findings of a role in hydrophobic protein degradation were intriguing, but the physiologic situation(s) when such a function becomes critical was not known. The current study very nicely addresses this issue and serves to identify a specific physiologic situation where Ubqln1 becomes critical, and provides proteomic validation on a cell-wide scale of observations previously made for only a couple of model proteins.

Specific major points that need to be addressed:

1) The experiment in Figure 8 should be better controlled in order for the authors to draw their intended conclusion. The best experiment would probably be to monitor eGFP translation rates when co-transfected with wild type versus signal-deleted ATP5G1. This would narrow the effect to non-imported ATP5G1 as opposed to an effect of simply over-expressing another protein in addition to eGFP. At the very least, some unrelated protein (e.g., mRFP) should be co-expressed in the control situation. As it currently stands, it appears the authors are comparing the situation where only eGFP is transfected, versus on where eGFP is co-transfected with the signal-disrupted ATP5G1. The observed effect on eGFP could be for reasons other than a specific inhibition of translation via a mis-targeted ATP5G1.

2) A potentially useful experiment would be to determine whether LPS stimulation combined with mitochondrial depolarization (e.g., with CCCP) shows a proliferation defect in Ubqln1 knockout cells. This would nicely establish that the key difference between LPS and BCR ligation is down to the latter's effect on mitochondrial depolarization. Because the central idea of this paper is that the double burden of mitochondrial biogenesis and attenuated import explains the phenotype of Ubqln1 deficient cells, the proposed experiment seems important.

3) Good characterization of the in vivo immune response to viral or antigen challenge. This is important to address the in vivo consequences of impaired BCR signaling and to support the notion that Ubiquilin 1 function is restricted to the BCR. Relatedly, is there an effect of Ubiquilin 1 loss on plasma cell differentiation due to an altered unfolded protein response? If so, would it also be relevant to the B-1 cell defect as the cells spontaneously secrete Ig?

4) The lack of a clear impact on mitochondrial function is surprising and weakens the mechanistic basis of the study. The authors should comment on this point.

---

## [Author Response]

Specific major points that need to be addressed:1) The experiment in Figure 8 should be better controlled in order for the authors to draw their intended conclusion. The best experiment would probably be to monitor eGFP translation rates when co-transfected with wild type versus signal-deleted ATP5G1. This would narrow the effect to non-imported ATP5G1 as opposed to an effect of simply over-expressing another protein in addition to eGFP. At the very least, some unrelated protein (e.g., mRFP) should be co-expressed in the control situation. As it currently stands, it appears the authors are comparing the situation where only eGFP is transfected, versus on where eGFP is co-transfected with the signal-disrupted ATP5G1. The observed effect on eGFP could be for reasons other than a specific inhibition of translation via a mis-targeted ATP5G1.

We agree with the reviewer that the experiment needs additional elaboration and controls. We performed several variations on the experiment to address these issues. The protocol for the experiment is quite complicated, unfortunately involving perturbation of the cells’ physiology by electroporation, by ficoll gradient fractionation, and by AHA labeling. Apparently because of these elements in the design of the experiments, which we have not yet been able to bypass, we see noise levels that are problematic. These will take considerable additional time to resolve, and we think it will fit best into a subsequent study. Therefore, we have decided to remove Figure 8. Note that we have removed and altered text in the Discussion, to reflect a more nuanced claim.

We have also expanded our discussion of the findings from Wrobel et al. and Wang et al., 2015 (1,2), in the fourth paragraph of the Discussion, to better explain the excellent prior literature on the link between mitochondrial protein accumulation and protein synthesis. These two back-to-back Nature papers provide good evidence that accumulation of mislocalized mitochondrial proteins is sufficient to inhibit cytosolic protein synthesis in yeast. In these reports, it was found that inhibition of mitochondrial protein import – either by genetic mutation or deletion of mitochondrial protein import machinery, or of genes involved in maintaining mitochondrial polarity – leads to a general inhibition of cytosolic protein synthesis as measured by metabolic labeling with S^35^ methionine. Wang et al. also quantified the proteome of yeast with collapsed mitochondrial polarity by mass spectrometry and found a specific accumulation of nucleus-encoded mitochondrial proteins, two of which had MTS sequences still attached.

We think that these findings support our hypothesis that the protein synthesis defect observed upon BCR stimulation in the absence of UBQLN1 likely results from the accumulation of mislocalized mitochondrial proteins. To improve on this line of inquiry, given the difficulties mentioned above and the high quality of the yeast literature, will require work at a depth that we consider beyond the scope of the current manuscript. With this in mind, we think that the removal of Figure 8 does not significantly alter the conclusions of our manuscript.

2) A potentially useful experiment would be to determine whether LPS stimulation combined with mitochondrial depolarization (e.g., with CCCP) shows a proliferation defect in Ubqln1 knockout cells. This would nicely establish that the key difference between LPS and BCR ligation is down to the latter's effect on mitochondrial depolarization. Because the central idea of this paper is that the double burden of mitochondrial biogenesis and attenuated import explains the phenotype of Ubqln1 deficient cells, the proposed experiment seems important.

We appreciate the reviewers’ point here and have performed experiments to address this concern. At first, we attempted to depolarize LPS-stimulated cells with the ETC uncoupler, CCCP. We found that at all doses that high enough to induce depolarization, the cells died within 24-48 hours, precluding analysis of effects on proliferation. We repeated the experiment using combinations of antimycin A and oligomycin in attempts to mitigate these toxicity problems, but found similar results.

As an alternative, we tested the effects ionomycin, which induces a calcium flux, on LPS stimulation. This stimulus also induced mitochondrial depolarization, in a way somewhat similar to BCR ligation, which also induces a prolonged increase in cytoplasmic Ca^2+^. Unfortunately, ionomycin strongly inhibited LPS-induced proliferation, even at doses that had little effect on mitochondrial polarization, and even in the presence of Ubqln1. Thus, although we find the reviewers’ suggested experiment would be helpful, we have not found a pharmacologic agent that allows testing of the hypothesis that mitochondrial depolarization alone is sufficient to inhibit cell cycle entry in the absence of *Ubqln1*. Each of the pharmacologic agents induces cytotoxicity at any dose that leads to mitochondrial depolarization.

3) Good characterization of the in vivo immune response to viral or antigen challenge. This is important to address the in vivo consequences of impaired BCR signaling and to support the notion that Ubiquilin 1 function is restricted to the BCR. Relatedly, is there an effect of Ubiquilin 1 loss on plasma cell differentiation due to an altered unfolded protein response? If so, would it also be relevant to the B-1 cell defect as the cells spontaneously secrete Ig?

We appreciate the reviewers’ requests, and have included some new data in Figure 3. We performed 3 independent in vivo immunizations of WT and -/- mice (n=5-7 mice per genotype) with a T-independent antigen (TNP-ficoll delivered IP) but did not find any Ubqln1-dependent defects in antibody production. Data from these experiments are now included in Figure 3 and discussed in the second paragraph of the Results subsection “Ubqln1 is required for cell cycle entry downstream of BCR stimulation”.

We are not certain why *Ubqln1*-deficient mice have normal antibody titers following TNP-ficoll immunization, but we speculate that there may be costimulatory signals in vivothat may compensate for the BCR-induced defect, as we observed in vitro(Figure 3). The defect in B1a cells that we have found suggests that it would be possible to design experiments that explored their particular importance in immune responses, but we believe this is essentially a new direction and would take significant additional time to address.

In addition to the immunization study we performed, in our 3-day in vitro proliferation experiments, we looked at the phenotype of surviving cells to see whether the absence of *Ubqln1* influenced the character as opposed to the proliferation rate of stimulated B cells, using Syndecan1 as a marker for plasmablast cells. Syndecan1 was not upregulated following BCR stimulation; however, following LPS stimulation, a large proportion of the cells became Syndecan1-high (not shown). We did not see any differences in the proportion of Syndecan1+ cells, or the level of Syndecan1 expression (not shown), in *Ubqln1^-/-^* cells following LPS stimulation.

We also examined stimulated cells for the induction of UPR via QPCR for HSPA5 and CHOP, but found no differences between WT and -/- cells (Author response image 1). Like plasma cell differentiation, we saw much more induction of the UPR upon LPS stimulation compared to BCR stimulation, and like previous reports, we saw more induction of HSPA5 (BiP) compared to CHOP (DDIT3) (3,4).

**Author response image 1. respfig1:** UPR gene expression following stimulation of primary B cells. Splenic B cells from WT or -/- mice were isolated and stimulated as in Figure 3 of manuscript. 24 hours after stimulation, cDNA samples were isolated for QPCR against HSPA5 and CHOP. Shown are mean ± SEM of biological triplicates.

Thus, *Ubqln1* appears to not be necessary for resolution of the UPR under these conditions. Instead, we find that *Ubqln1* is necessary to resolve a stress downstream of signaling that induces mitochondrial depolarization.

4) The lack of a clear impact on mitochondrial function is surprising and weakens the mechanistic basis of the study. The authors should comment on this point.

We, too, were originally surprised to see no effect of *Ubqln1* loss on mitochondrial function in B cells, and we examined the mitochondria quite exhaustively in our attempts to identify a Ubqln-dependent defect. However, our results are consistent with the results from Itakura et al. which showed no change in mitochondrial morphology (Figure S3E of their study) or expression levels of respiratory chain complex proteins (Figure S4A of their study) upon deletion of Ubqlns 1, 2, and 4.

Our hypothesis is that these mitochondrially-targeted proteins are made in excess even in cases of mitochondrial biogenesis, and some copies of those excessively synthesized proteins fail mitochondrial insertion. Those ‘rejected’ proteins require UBQLN1 for their degradation, while the mitochondria remain capable of functioning normally with the proportion of those proteins that do get properly inserted. This point is made in the third paragraph of the Discussion, as requested.References:

1) Wrobel L, Topf U, Bragoszewski P, Wiese S, Sztolsztener ME, Oeljeklaus S, Varabyova A, Lirski M, Chroscicki P, Mroczek S, Januszewicz E, Dziembowski A, Koblowska M, Warscheid B, Chacinska A. Mistargeted mitochondrial proteins activate a proteostatic response in the cytosol. Nature. 2015 Aug 5;524(7566):485–8.

2) Wang X, Chen XJ. A cytosolic network suppressing mitochondria-mediated proteostatic stress and cell death. Nature. Nature Research; 2015 Aug 27;524(7566):481–4.

3) Gass JN, Gifford NM, Brewer JW. Activation of an unfolded protein response during differentiation of antibody-secreting B cells. Journal of Biological Chemistry. American Society for Biochemistry and Molecular Biology; 2002 Dec 13;277(50):49047–54.

4) Ma Y, Shimizu Y, Mann MJ, Jin Y, Hendershot LM. Plasma cell differentiation initiates a limited ER stress response by specifically suppressing the PERK-dependent branch of the unfolded protein response. Cell Stress Chaperones. 2010 May;15(3):281–93. PMCID: PMC2866998.